# The autophagy adaptor NDP52 and the FIP200 coiled-coil allosterically activate ULK1 complex membrane recruitment

**Xiaoshan Shi**[1†], **Chunmei Chang**[1†], **Adam L Yokom**[1†], **Liv E Jensen**[1], **James H Hurley**[1,2]*

[1]Department of Molecular and Cell Biology and California Institute for Quantitative Biosciences, University of California, Berkeley, Berkeley, United States; [2]Molecular Biophysics and Integrated Bioimaging Division, Lawrence Berkeley National Laboratory, Berkeley, United States

**Abstract** The selective autophagy pathways of xenophagy and mitophagy are initiated when the adaptor NDP52 recruits the ULK1 complex to autophagic cargo. Hydrogen-deuterium exchange coupled to mass spectrometry (HDX-MS) was used to map the membrane and NDP52 binding sites of the ULK1 complex to unique regions of the coiled coil of the FIP200 subunit. Electron microscopy of the full-length ULK1 complex shows that the FIP200 coiled coil projects away from the crescent-shaped FIP200 N-terminal domain dimer. NDP52 allosterically stimulates membrane-binding by FIP200 and the ULK1 complex by promoting a more dynamic conformation of the membrane-binding portion of the FIP200 coiled coil. Giant unilamellar vesicle (GUV) reconstitution confirmed that membrane recruitment by the ULK1 complex is triggered by NDP52 engagement. These data reveal how the allosteric linkage between NDP52 and the ULK1 complex could drive the first membrane recruitment event of phagophore biogenesis in xenophagy and mitophagy.

*For correspondence:
jimhurley@berkeley.edu

†These authors contributed equally to this work

## Introduction

Autophagy is a cellular process that maintains homeostasis via clearance of cellular components, degradation of damaged organelles and protection against viral and bacterial invasion (*Boya et al., 2013*). Autophagy entails the de novo formation of a double membrane structure (termed the isolation membrane), maturation into a sealed autophagosome, and delivery of substrates to the lysosome for degradation (*Melia et al., 2020*; *Mercer et al., 2018*). Macroautophagy and selective autophagy are two distinct modes of autophagy that differ in their initiation mechanisms and cargo specificity (*Yu et al., 2018*; *Zaffagnini and Martens, 2016*). During cellular starvation conditions, macroautophagy functions by the initial formation of an isolation membrane at ER exit sites and expands (*Graef et al., 2013*; *Suzuki et al., 2013*), engulfing cytoplasmic debris in a nonspecific manner. Selective autophagy functions via a set of autophagy adaptor proteins, which bind cargos directly or via polyubiquitin chains (*Zaffagnini and Martens, 2016*). The isolation membrane forms on the labeled cargo and tightly engulfs the cargo during maturation. A complex cascade of autophagy (ATG) related proteins, including the ULK1 complex, Class III PI3K complex, WIPIs, ATG12-5-16, LC3, ATG9, cargo adaptors, and others are required for mammalian autophagy (*Bento et al., 2016*; *Hurley and Young, 2017*). Through numerous studies, selective autophagy pathways for mitochondria (mitophagy), aggregates (aggrephagy), lysosomes (lysophagy), endoplasmic reticulum (ER-phagy), bacteria (xenophagy) and other cargos have been discovered (*Zaffagnini and Martens, 2016*). Correspondingly, a set of adaptors have been identified which function to decorate cargos for degradation. Nuclear domain 10 protein 52 (NDP52), optineurin, CCPG1, TAX1BP1 and p62 represent a subset of such adaptors (*Kirkin and Rogov, 2019*).

NDP52, also known as Calcium Binding And Coiled-Coil Domain 2 (CALCOCO2), is a multifunctional autophagy adaptor that is crucial for both xenophagy and mitophagy (*Lazarou et al., 2015*; *von Muhlinen et al., 2010*). Knock out studies of NDP52 show a severe impairment of autophagic protection against *Salmonella* invasion (*Verlhac et al., 2015*) and diminished recruitment of the ULK1 complex during mitophagy (*Lazarou et al., 2015*). The N-terminal SKICH domain of NDP52 forms a network of interactions in xenophagy and mitophagy pathways (*Ravenhill et al., 2019*; *Till et al., 2013*). Recent work has shown the direct interaction of NDP52 with FIP200, a protein in the autophagy initiation ULK1 complex (*Ravenhill et al., 2019*; *Vargas et al., 2019*). This interaction is crucial for recruitment of ULK1 to initial phagophore sites, ultimately leading to the engulfment and degradation of selected cargo. NDP52 contains a LIR motif which is required for its interaction with LC3, a hallmark of the autophagosome membrane. Lastly, NDP52 has two domains that recognize cargo; a GALB1 domain, which interacts with galectin-8, and a UBZ domain that recognizes polyubiquitin conjugated to the bacterial surface (*Thurston et al., 2009*).

The ULK1 complex is comprised of the ULK1 kinase, focal adhesion kinase family interacting protein of 200 kDa (FIP200), autophagy-related protein 13 (ATG13), and ATG101 (*Ganley et al., 2009*; *Hosokawa et al., 2009*; *Jung et al., 2009*; *Lin and Hurley, 2016*; *Mercer et al., 2009*). ULK1 and 2 are interchangeable threonine/serine kinases which function as the catalytic subunit of the ULK1 complex. The N-terminal kinase domain is followed by a ~ 550 residue intrinsically disordered region (IDR) and a C-terminal early autophagy tethering (EAT) domain which recruits ULK1 to the rest of the complex via ATG13 (*Alers et al., 2014*). This network of interactions is conserved in the yeast homolog Atg1 complex (*Fujioka et al., 2014*; *Stjepanovic et al., 2014*). FIP200 is a 1594 residue protein which contains a dimeric N-terminal scaffolding domain (*Shi et al., 2020*), an IDR, a coiled-coil (CC) region and a C-terminal 'Claw' domain (*Turco et al., 2019*). The C-shaped N-terminal domain dimer of FIP200 scaffolds assembly of the ULK1 complex while the C-terminal Claw domain interacts with cargo adaptors, including NDP52, p62 and CCPG1 (*Ravenhill et al., 2019*; *Turco et al., 2019*; *Vargas et al., 2019*). ATG13/101 form a HORMA dimer which interacts with PI3K and is required for ULK1 complex function and assembly (*Qi et al., 2015*; *Suzuki et al., 2015*). Little is known about the structure and function of the massive CC domain of FIP200, beyond the crystal structure of its very C-terminal 100 residues (*Turco et al., 2019*) and the identification of an NDP52 binding site between residues 1351–1441 (*Ravenhill et al., 2019*).

The ULK1 complex is rapidly recruited to invading bacteria and damaged mitochondria following their ubiquitination via its interaction with NDP52 (*Ravenhill et al., 2019*; *Turco et al., 2020*; *Vargas et al., 2019*). As the most upstream of the autophagy core complexes, ULK1 complex recruitment is pivotal to subsequent events in autophagosome biogenesis. The nature of the structural changes in the ULK1 complex following NDP52 engagement and how those changes influence recruitment of membranes and downstream autophagy components is, however, almost entirely unknown.

Despite progress in crystallizing substructures of the ULK1 complex (*Lin and Hurley, 2016*) and an intermediate resolution cryo-EM structure of the FIP200 NTD (*Shi et al., 2020*), progress on the intact ULK1 complex containing full-length FIP200 has been very challenging. FIP200 is 1594 amino acid residues in length, and apart from the NTD and C-terminal Claw, consists of an intrinsically disordered region (IDR) and a massive 710-residue CC. These features have made full-length FIP200 and the intact ULK1 complex exceptionally difficult for biochemical and structural studies. Here we demonstrate that purified full length FIP200 forms an extended and dynamic structure, which scaffolds the ULK1 complex at its dimeric C-shaped N-terminal domain. Binding of NDP52 to full length FIP200 and the ULK1 complex does not change the conformation of the N-terminal domain, but alters that of a large portion of the FIP200 CC domain. This interaction increases the membrane affinity of FIP200. NDP52 induces robust recruitment of the ULK1 complex to giant unilamellar vesicles (GUVs) by exposing a membrane binding site within the CC domain of FIP200.

## Results

### Full length FIP200 forms an elongated scaffold for ULK1 and adaptor binding

We sought to investigate the structure of purified full-length FIP200. To express and purify full-length FIP200, a dual tag construct was developed. Wild-type FIP200 was expressed with a glutathione S-transferase (GST) tag fused to the N-terminus and a maltose binding protein (MBP) tag on the C terminus. After affinity purification, full-length GST-FIP200-MBP was characterized using negative stain electron microscopy (NSEM) (*Figure 1A*, *Figure 1—figure supplement 1A*). Full length FIP200 single particles showed an extended density with a C-shaped density at one end, corresponding to the dimeric FIP200NTD previously resolved (*Shi et al., 2020*). Density for the GST tags can be seen in the center of the FIP200 NTD dimer (*Figure 1—figure supplement 1B*). This N-terminal density is followed by a long meandering density which we assigned as the CC domain (residues 790–1500). The IDR region (640-790) was not visualized. The CC domain of FIP200 has segments of predicted dimeric coiled-coils and linker regions. Features visualized at the C-terminus of FIP200 correspond to density for two MBP tags (40 kDa each) and the Claw dimer (22 kDa) (*Figure 1—figure supplement 1C*).

The CC was observed in a variety of conformations with varying lengths and curvatures. FIP200-CC domain comprises 710 residues which, if uninterrupted, would span about 105 nm. The contour lengths of the CC region in particles analyzed by NSEM ranged from 37 nm to 107 nm, with a mean length of 79 nm (*Figure 1B*). The presence of some particles with contour lengths far below the predicted maximum suggests that the CC is capable of folding back onto itself. The variability of the total length corresponds to a range of end-to-end distances distributed from 32 nm to 94 nm with a mean of 69 nm (*Figure 1B*). It is clear that the FIP200 N-terminus is separated from the C-terminus Claw domain. This spatially separates the site of ULK1 assembly at the NTD and the binding site of the cargo adaptors NDP52 near the C-terminal end of the CC (*Ravenhill et al., 2019*; *Vargas et al., 2019*) and p62 at the Claw domain C-terminal to the CC (*Turco et al., 2019*).

To investigate how the ULK1 complex assembles on full-length FIP200, we employed hydrogen-deuterium exchange mass spectrometry analysis (HDX-MS) on FIP200 alone and the full-length ULK1 complex. Purified ULK1 complex was active with respect to the synthetic substrate, ULKtide (*Figure 1—figure supplement 2A*). Difference maps for H/D exchange showed a clear protection pattern in the FIP200NTD, similar to our previous results on the FIP200 NTD alone (*Shi et al., 2020*). Specifically, peptides spanning residues 75–80, 157–165, 313–324, 350–356, 435–469 and 532–541 showed increases in protection of >10% after exchanging for 60 s (*Figure 1C*, *Figure 1—figure supplement 2B*). No large changes (>10%) were observed in FIP200 CC or Claw. This implies that assembly of the ULK1 complex with full length FIP200 does not affect the dynamics of the cargo-recognizing CC and Claw. Thus, FIP200 is a modular protein whose NTD engages with the rest of the ULK1 complex and whose CC and Claw engage independently with cargo adaptors.

### ULK1 complex engages NDP52 only via FIP200

Next, we used HDX-MS to systematically analyze how the H/D exchange rate of the ULK1 complex changed in the presence of NDP52. To validate the ULK1 complex sample, the complex was expressed with fluorescent GFP tags on ATG13 and ATG101 and purified. Labeled ULK1 complex was mixed with GST-NDP52, GST-4xUb and glutathione beads. The fluorescence signal was quantified for each reaction condition (*Figure 1—figure supplement 2C,D*). The ULK1 complex was specifically pulled down by GST-NDP52, but not by GST-4xUb or GSH beads alone. This confirmed that the ULK1 complex recognizes cargo adaptors but not poly-ubiquitin chains directly in our in vitro system.

In the presence of NDP52, no large changes (<-10% or >10%) were identified in peptides from ATG13, ATG101, ULK1 or the FIP200NTD (*Figure 2A–D*, *Figure 2—figure supplement 1*, all proteins at 2 μM, except FIP200 at 4 μM and NDP52 at 16 μM). A lack of HDX difference suggests that NDP52 does not directly or allosterically modulate these regions upon binding to the FIP200 CTD. However, our analysis showed three regions of the CC domain of FIP200 with significant protection in the presence of NDP52. Peptides of residues 1290–1320, 1339–1392 and 1462–1470 showed a decrease of deuterium exchange of >10% (*Figure 2D*, *Figure 2—figure supplement 2A*). Residues

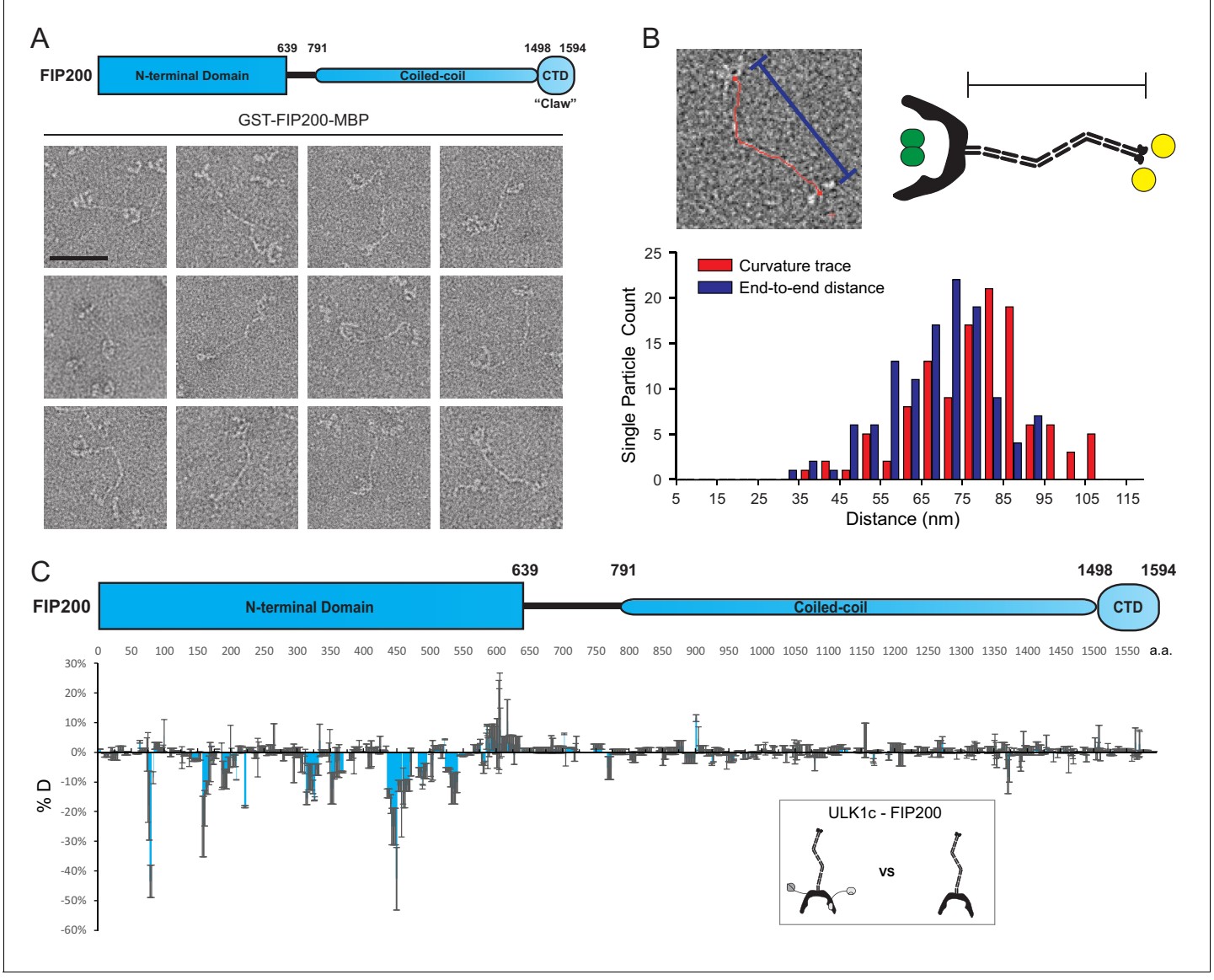

**Figure 1.** EM and HDX-MS of full-length FIP200. (**A**) Negative stain EM single particles of full length FIP200 alone. Scale bar 50 nm. (**B**) Histogram of FIP200 path length and end-to-end distances. (**C**) Difference of Hydrogen Deuterium Exchange percentages of the FIP200 alone vs FIP200:ATG13: ATG101:ULK1 at 60 s time point. All values are mean (Blue) ± SD (Grey). N = 3 replicates.

The online version of this article includes the following source data and figure supplement(s) for figure 1:

**Figure supplement 1.** EM of full-length FIP200.

**Figure supplement 2.** Purified ULK1 complex is functional.

**Figure supplement 2—source data 1.** Source data for graphs in *Figure 1—figure supplement 2*.

in the 1339–1392 region (L1371, L1378, L1385, and L1392) corresponds to the previously proposed as a binding site of NDP52 (*Ravenhill et al., 2019*; *Vargas et al., 2019*). Residues 1290–1320 and 1339–1392 display a higher level of protection than 1462–1470. Unexpectedly, an extensive region of the CC N-terminal to the NDP52 binding site, spanning FIP200 residues 800–1250, especially 935–1073, had increased H/D exchange rate after NDP52 binding (*Figure 2D*, *Figure 2—figure supplement 2A*, *Supplementary file 1*- Table S1). These changes suggest that the N-terminal part of the CC is destabilized upon NDP52 binding.

To confirm the conclusion from HDX-MS that ULK1, ATG13 and ATG101 are not involved in NDP52 binding, fluorescent FIP200 alone and the complete ULK1 complex were compared in a GST pulldown assay with GST-NDP52 as bait (*Figure 2E*). FIP200 alone and the full ULK1 complex bound

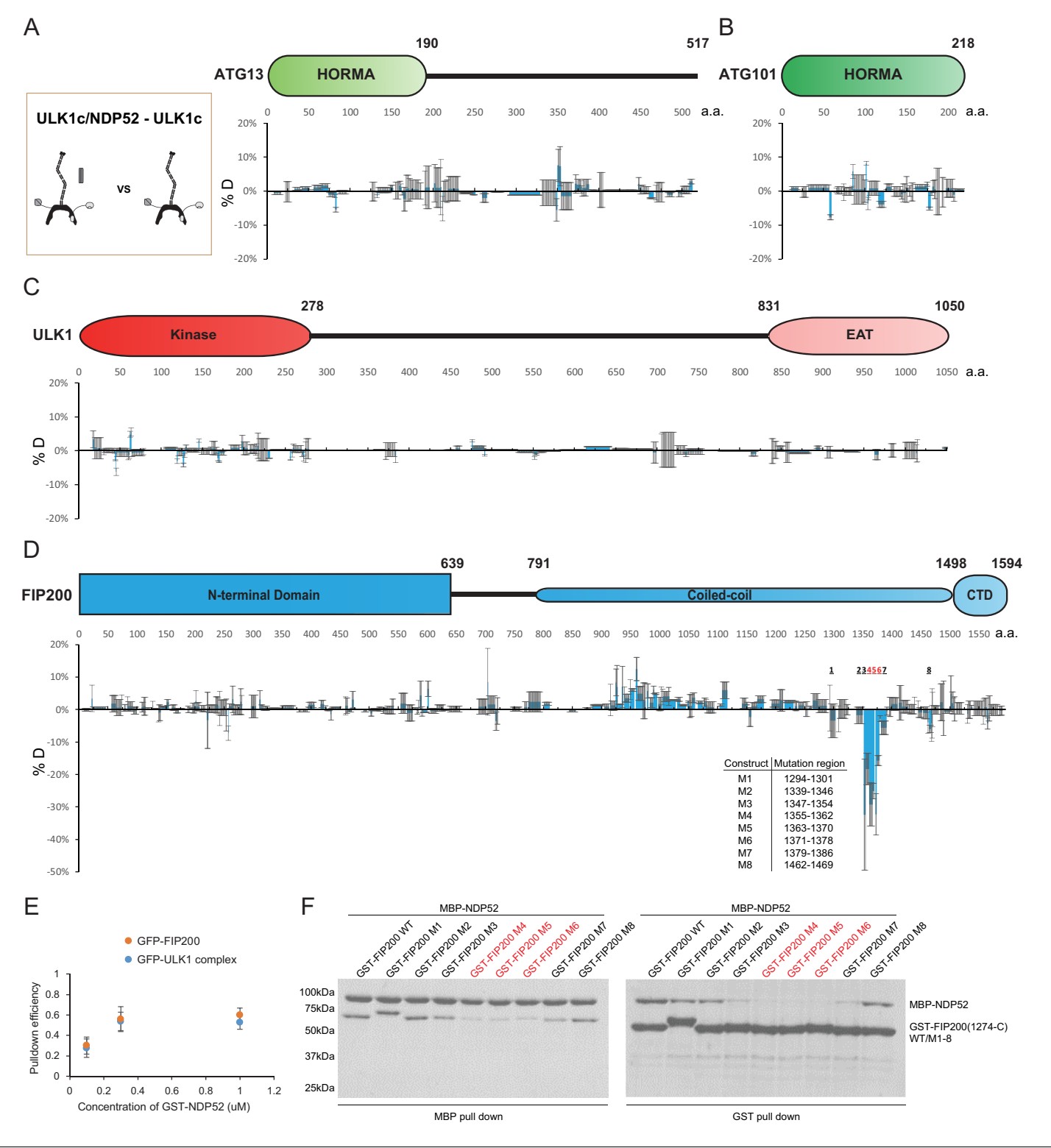

**Figure 2.** HDX-MS mapping of NDP52 interactions with the ULK1 complex. (**A–D**) Difference of Hydrogen Deuterium Exchange percentages of the ATG13 (**A**), ATG101 (**B**), ULK1 (**C**) and FIP200 (**D**) in ULK1 complex vs in ULK1 complex with NDP52 at the 60 s time point. All values are mean (Blue) ± SD (Grey). N = 3 replicates. (**E**) Pull-down efficiency of GFP-tagged wild type ULK1 complex or GFP-FIP200 by glutathione sepharose beads coated with different concentrations of GST-NDP52 as baits. All values are mean ± SD. N = 4 biological replicates. (**F**) Pull-down assays of mutant

*Figure 2 continued on next page*

eLife Research article

Biochemistry and Chemical Biology | Cell Biology

*Figure 2 continued*

FIP200 constructs (M1–M8) and wild type with NDP52. Both GSH and Amylose resin were used to pull down GST-FIP200(1274 C):MBP-NDP52 complex from lysate of overexpressing HEK cells. The pull-down results were visualized by SDS-PAGE and Coomassie blue staining.

The online version of this article includes the following source data and figure supplement(s) for figure 2:

**Source data 1.** Source data for graph in *Figure 2E*.
**Figure supplement 1.** HDX-MS analysis of the interaction of the ULK1 complex with NDP52.
**Figure supplement 2.** The coiled-coil domain of FIP200 is involved in binding with NDP52 and membranes, but not its stability.

to the same extent to 0.1, 0.3 and 1 uM of GST-NDP52. To further investigate which regions of FIP200 bind directly to NDP52, we mutated eight regions identified by our HDX-MS analysis (*Figure 2D*). Regions selected for mutation were replaced by a poly Gly-Ser sequence of equal length to the wild-type region. GST pulldown assays showed that the expression of FIP200 was unaffected by the Gly-Ser mutations (*Figure 2F*, right). In addition, both GST and MBP pulldown assays confirmed that mutation of region 4 (1355–1362), 5 (1363–1370) and 6 (1371–1378) in FIP200 impaired the interaction with NDP52, while the rest showed small decreases in binding affinity (region 3 and 7) or minimal (region 1, 2 and 8) effects (*Figure 2F*).

## FIP200 CC interacts with membranes

The finding that NDP52 increased the H/D exchange rate of a subset of the coiled-coil (800–1250) region of FIP200 stimulated our curiosity about the physiological function of this CC region. We thus used HDX-MS to systematically analyze if the ULK1 complex interacts with SUVs containing PI lipids. No large changes (<-10% or >10%) were identified in peptides from ATG13 or ATG101 which suggests that PI containing membranes do not directly or allosterically modulate these proteins of the ULK1 complex (*Figure 3A,B*). In our assay, two regions of FIP200 showed significant protection. Residues within the N-terminal domain (539-564) showed moderate protection ~10%. However, the strongest protection profile (5–45%) was seen across an extended portion of the CC domain, covering residues 844–1073 (*Figure 3C*, *Figure 2—figure supplement 2B*).

To refine the mapping of the membrane binding site within FIP200, we compared truncations of the FIP200 in a liposome sedimentation assay. NTD (N-640) and CTD (636-C) FIP200 proteins were mixed with SUVs before separation into pellet and supernatant fractions (*Figure 3D*). The FIP200 CTD (636-C) containing the CC domain, but not FIP200 NTD, co-precipitated with membranes. This was most prominent in SUVs containing an acidic lipid mixture of PS/PI. Furthermore, a smaller amount of the FIP200 CTD was observed in the pellet of SUVs containing PC/PE. The interaction between the CC domain of FIP200 and membranes raises the possibility that NDP52 binding at the C-terminal CC domain could trigger destabilization or opening of the N-terminal CC domain, increasing exposure of hydrophobic residues and so stimulating membrane binding to the ULK1 complex.

## Reconstitution of NDP52-stimulated membrane binding of the ULK1 complex

To investigate how NDP52 binding effects membrane recruitment of the ULK1 complex, we reconstituted this event on GUVs with an ER-like lipid composition. FIP200 alone was only minimally recruited to GUVs (*Figure 4A,B*). Additionally, minimal GUV binding was observed in the presence of His-tagged NDP52. GST-tagged NDP52 showed a small increase in membrane binding (*Figure 4A,B*). Given that the GST tag forms a native dimer, this suggested the oligomeric state of NDP52 was crucial for the activation of FIP200 membrane binding.

To test how NDP52 oligomeric state effects FIP200 recruitment we included GST-tagged linear tetra-ubiquitin, which mimics poly-ubiquitinated protein substrates and induces the oligomerization of NDP52 in vitro. No recruitment of FIP200 was detected to GUVs in the presence of GST-4xUb alone, consistent with the lack of a direct interaction between FIP200 and Ub (*Figure 1—figure supplement 2D*). In the presence of GST-4xUb and NDP52, FIP200 rapidly coalesced into clusters and was robustly recruited to the GUV surface (*Figure 4A,B*). These clusters began to form within 5 min of mixing (*Figure 4A,B*). After 35 min, FIP200 was clustered into large, continuous domains on the GUVs (*Figure 4A,B*).

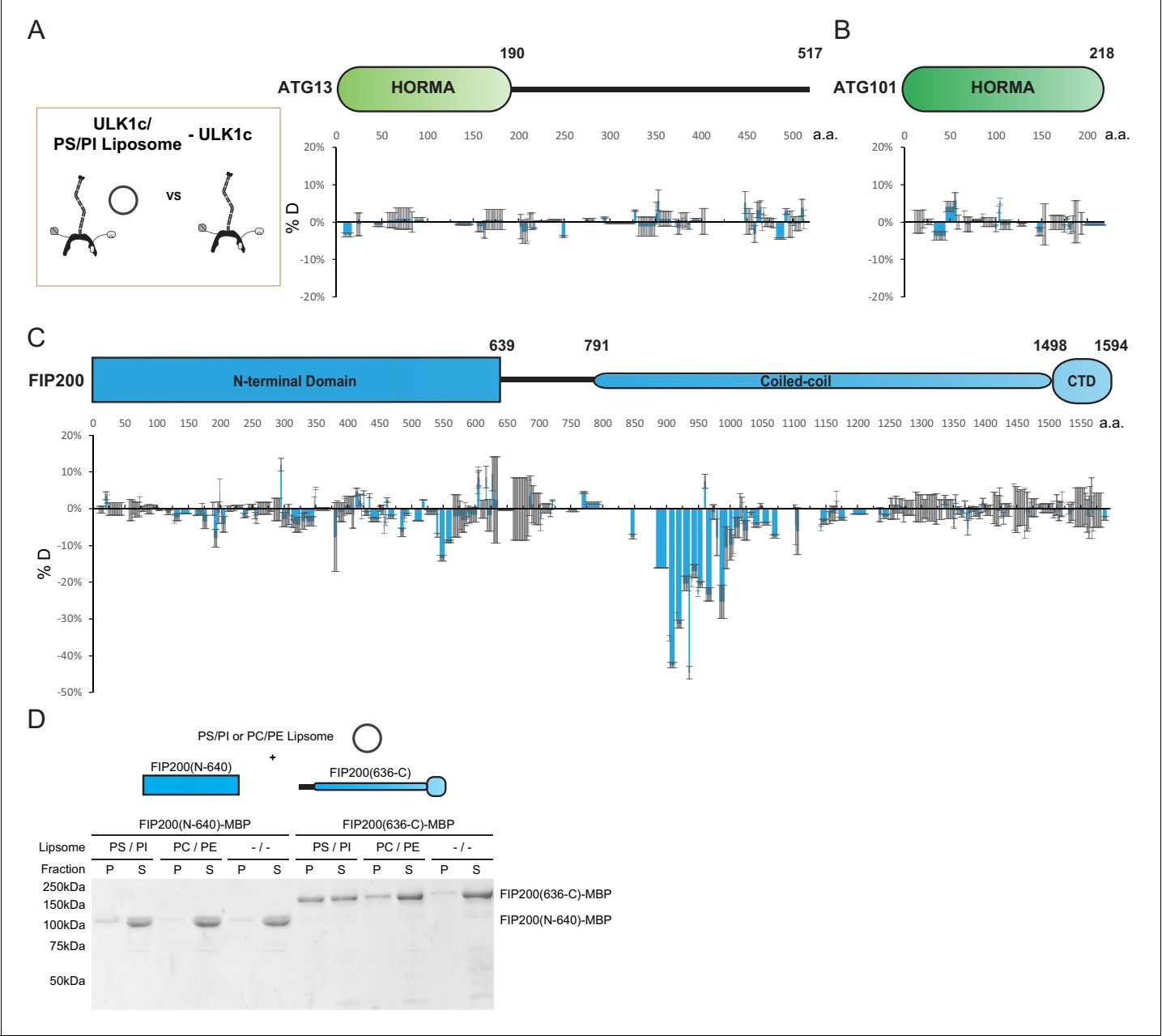

**Figure 3.** HDX-MS mapping of membrane interactions with the ULK1 complex. (**A–C**) Difference of Hydrogen Deuterium Exchange percentages of the ATG13 (**A**), ATG101 (**B**) and FIP200 (**C**) in ULK1 complex vs in ULK1 complex with POPS/POPI SUV at the 60 s time point. All values are mean (Blue) ± SD (Grey). N = 3 replicates. (**D**) Liposome sedimentation assay of FIP200 truncations alone with POPS/POPI and POPC/POPE SUVs. Results were visualized by SDS-PAGE and Coomassie blue staining with the supernatant fractions (S) and pellet fractions (P).

We then tested the membrane recruitment of the entire ULK1 complex under equivalent conditions. Similarly to FIP200 alone, full-length ULK1 complex formed clusters with NDP52 and poly-ubiquitin but not in their absence (*Figure 4C,D*). Recruitment of the ULK1 complex was stronger with the dimeric GST-NDP52 than the monomeric His-NDP52. Together, these data show that oligo-merized NDP52 activates membrane recruitment of the ULK1 complex. Furthermore, the higher order assembly of NDP52 via poly-ubiquitin stimulates membrane binding.

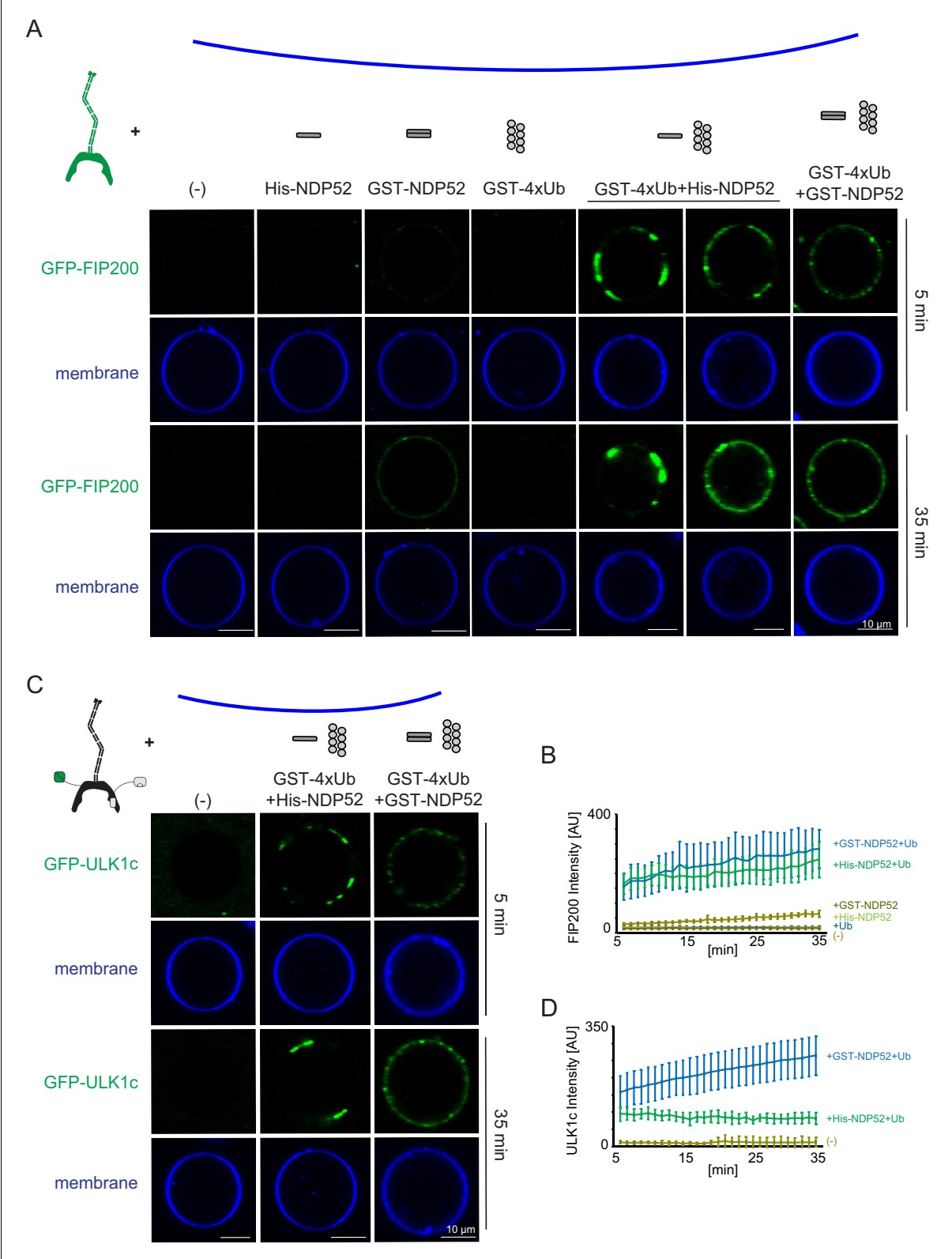

**Figure 4.** Reconstitution of NDP52-stimulated membrane binding of the ULK1 complex. The schematic drawing illustrates the reaction setting. Colors indicate fluorescent protein fused components. Components in gray are not labeled but are present in the reaction mix. (**A**) Representative confocal micrographs showing the membrane recruitment of GFP-FIP200. GFP-FIP200 mixed with His-NDP52 or GST-NDP52 was incubated with GUVs, either in the absence or presence of GST-4xUb at room temperature. GFP-FIP200 alone or mixed with GST-4xUb was also incubated with GUVs at room

*Figure 4 continued on next page*

*Figure 4 continued*
temperature as controls. Images taken at indicated time points were shown. Scale bars, 10 μm. (B) Quantitation of the kinetics of FIP200 recruitment to the membrane from individual GUV tracing in A (means ± SDs; N = 51 (-); 51 (His-NDP52); 48 (GST-NDP52); 53 (GST-4xUb); 72 (GST-4xUb+ His-NDP52); 57 (GST-4xUb+ GST-NDP52)). (C) Representative confocal micrographs showing the membrane recruitment of GFP-ULK1 complex. GFP-ULK1 complex alone or mixed with His-NDP52 or GST-NDP52 in the presence of GST-4xUb was incubated with GUVs at room temperature. Images taken at indicated time points were shown. Scale bars, 10 μm. (D) Quantitation of the kinetics of ULK1 complex recruitment to the membrane from individual GUV tracing in C (means ± SDs; N = 54 (-); 47 (GST-4xUb+ His-NDP52); 49 (GST-4xUb+ GST-NDP52)).
The online version of this article includes the following source data for figure 4:

**Source data 1.** Source data for GUV image quantitation data in *Figure 4*.

## NDP52 binding is required for the membrane recruitment of ULK1

To validate that the middle region of FIP200 (residues 790–1050) was the membrane binding site of ULK1 complex as suggested by HDX-MS, we generated FIP200 constructs with the middle region (residues 790–1050) deleted (FIP200$^{\Delta MR}$). The NDP52 binding region from HDX-MS, residues 1363–1370 was also converted to poly Gly-Ser (FIP200$^{\Delta NDP52}$) in a separate mutation. Neither mutation altered the stability of FIP200 on the basis that both mutants yielded protein at the same level as wild type (*Figure 2—figure supplement 2C*). A microscopy-based bead assay was used to confirm that FIP200$^{\Delta MR}$ still bound to NDP52 (*Figure 5A,B*). WT and FIP200$^{\Delta MR}$ ULK1 complex were both robustly recruited to GST-NDP52 beads (*Figure 5A,B*). FIP200$^{\Delta NDP52}$ ULK1 was not recruited to the GST-NDP52 beads (*Figure 5A,B*), which is consistent with our previous pulldown data (*Figure 2F*).

We proceeded to test the membrane recruitment of ULK1 complexes containing these two mutants. WT ULK1 complex was recruited strongly in the presence of GST-NDP52 and GST-4xUb (*Figure 5C,D*). FIP200$^{\Delta MR}$ ULK1 complex bound more weakly to GUVs (*Figure 5C,D*), whereas FIP200$^{\Delta NDP52}$ ULK1 complex completely abolished membrane recruitment (*Figure 5C,D*). These data show that the NDP52-FIP200 interaction is essential for membrane recruitment. The CC membrane binding site is also important for binding, but residual binding still occurs in when the CC membrane binding site is removed. We tested whether NDP52 itself could bind membranes. NDP52 was recruited to the GUV membranes either when it was tagged with GST or upon the addition of poly-ubiquitin (*Figure 5—figure supplement 1*). From this finding, we concluded that the residual binding to GUVs is due to membrane binding by NDP52, not to additional membrane binding sites on the ULK1 complex.

## Discussion

The year 2019 saw a paradigm shift in the initiation of selective autophagy (*Melia et al., 2020*). Recognition of p62 (*Turco et al., 2019*) and NDP52 (*Ravenhill et al., 2019*; *Vargas et al., 2019*) by FIP200 was shown to the key triggering event responsible for recruiting the ULK1 complex to cargo. The binding sites of p62 and NDP52 were mapped to the Claw (*Turco et al., 2019*) and CC C-terminus (*Ravenhill et al., 2019*) respectively, and the structure of the Claw and a small part of the CC were determined (*Turco et al., 2019*). All of the many downstream events remained unclear, however, as did the structure of the great majority of FIP200. Here we obtained the first structural view of full-length FIP200 and the intact ULK1 complex. In the context of the structure, we formulated a model for the allosteric activation by NDP52 of the next step following ULK1 complex recruitment to cargo, namely membrane recruitment.

In a related study, we recently showed that the N-terminal domain (NTD) of FIP200, consisting of the first 640 amino acids, comprised a C-shaped dimeric hub for assembly of the rest of the ULK1 complex. A single molecule of ATG13 was shown to bind to the NTD dimer. ATG13 in turn contacts ATG101 and ULK1 itself. The ULK1 complex is unusually loosely organized, lacking extensive direct contacts between FIP200, ULK1, and ATG101. Biochemical and structural studies of the 1594-residue FIP200 have been exceptionally challenging, and it has only now been possible to visualize the intact molecule. The C-termini of the two FIP200 NTDs in the C-shaped dimer is located near the center. We have now confirmed that the CC projects away from the center of the NTD. Thus, the symmetry of the NTD matches the symmetry of the FIP200 CC, which is parallel and a dimer (*Turco et al., 2019*). FIP200 CC has a modal end-to-end distance of ~75 nm and is partially flexible, making it ideally suited to its function of connecting cargo to other components of the ULK1

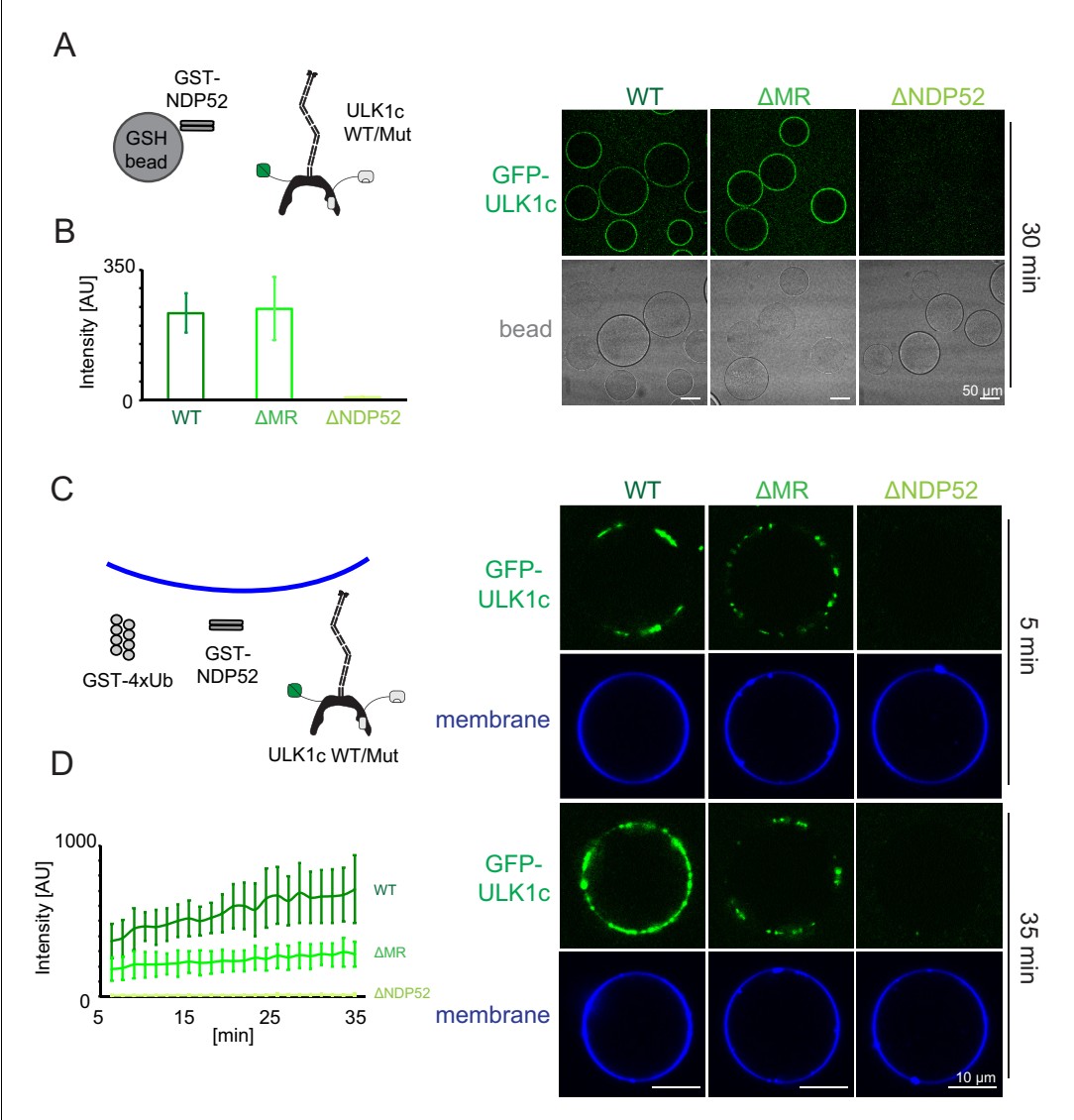

**Figure 5.** NDP52 allosterically activates membrane association of the ULK1 complex. (A) Microscopy-based bead protein interaction assay with glutathione sepharose beads coated with GST-NDP52 as baits and incubated with GFP-tagged wild type ULK1 complex or mutant as prey. Representative confocal micrographs are shown. Scale bars, 50 μm. (B) Quantification of the GFP-ULK1 complex signal intensity measured on glutathione sepharose beads coated with GST-NDP52 (means ± SDs; N = 20). (C) Representative confocal micrographs showing the membrane recruitment of GFP-ULK1 complex. GFP-tagged wild type ULK1 complex or mutant was mixed with GUVs in the presence of GST-NDP52 and GST-4xUb at room temperature. Images taken at indicated time points were shown. Scale bars, 10 μm. (D) Quantitation of the kinetics of ULK1 complex recruitment to the membrane from individual GUV tracing in A (means ± SDs; N = 22 (WT); 25 (ΔMR); 22 (ΔNDP52)).

The online version of this article includes the following source data and figure supplement(s) for figure 5:

**Source data 1.** Source data for GUV image quantitation data in *Figure 5*.

**Figure supplement 1.** Membrane binding of NDP52.

complex and to membranes over substantial distances, much as seen with other coiled-coil based tethers such as the endosomal tether EEA1 (*Murray et al., 2016*) and the Golgi tether GCC185 (*Cheung et al., 2015*).

The data presented here suggest that the membrane binding region (MR) in the N-terminal portion of FIP200 CC is the principal, although not sole, membrane binding site responsible for the first ULK1 complex membrane recruitment event in autophagy. NDP52 and, most probably, the FIP200 NTD also contribute to membrane interactions. *Figure 6* depicts the series of events that these data

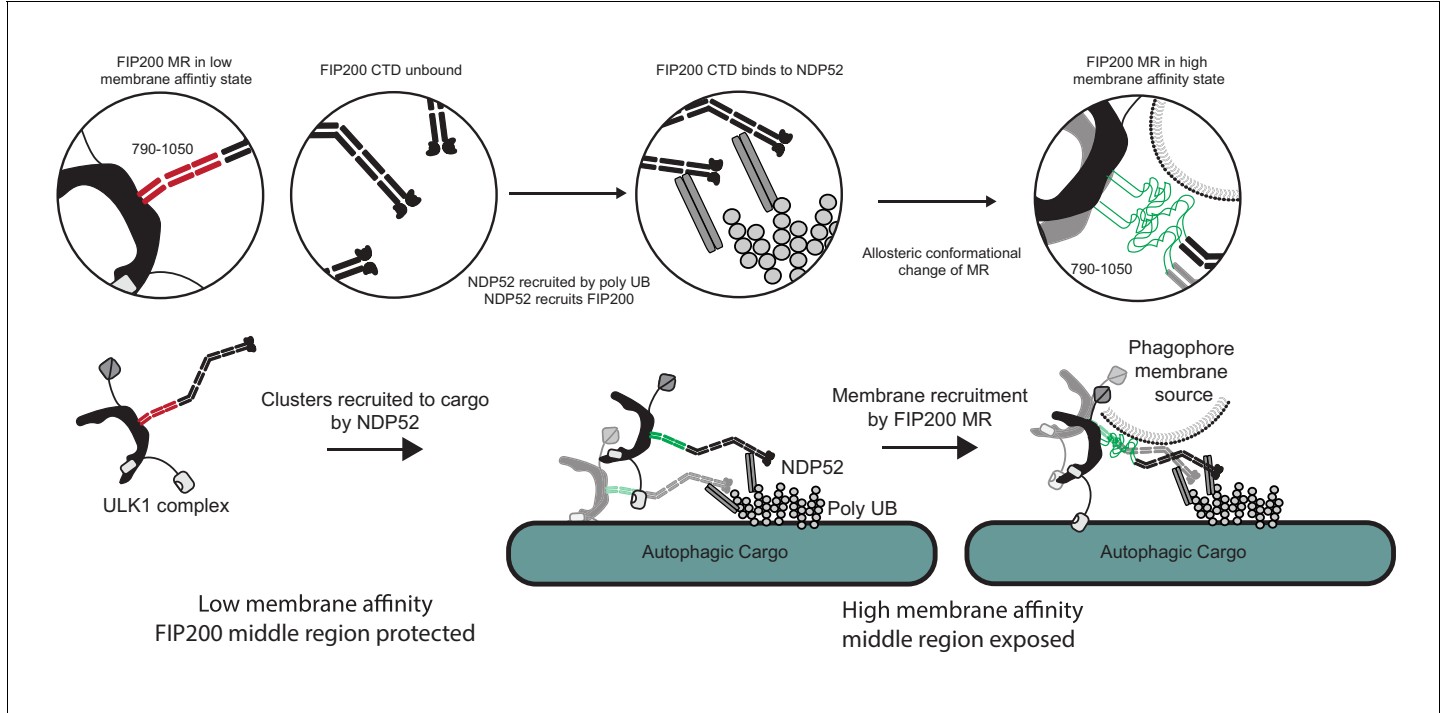

**Figure 6.** Model for ULK1 complex membrane recruitment. Before engagement with NDP52, the FIP200 middle region (790–1050) forms a stable coiled-coil in a low membrane affinity state and the CTD and 'Claw' domains are unbound. Initially, NDP52 is recruited by interaction with ubiquitin to autophagic cargo in both xenophagy and mitophagy. FIP200 CTD binds directly to NDP52, driving clustering of FIP200 along with an allosteric conformational change in the MR region. These clusters form a hub for ULK1 auto-transphosphorylation and a site for initial recruitment of phagophore membranes.

lead us to believe underlie membrane recruitment downstream of NDP52-ubiquitin engagement. In solution, FIP200 exists as isolated dimers with the MR region in a helical conformation that sequesters the hydrophobic residues. When ubiquitin is conjugated to and presented on the surface of damaged mitochondria, invading bacteria, or other substrates, NDP52 is recruited to these sites by directly binding to the clustered ubiquitin on their surfaces (*Thurston et al., 2009*; *Thurston et al., 2012*; *Xie et al., 2015*). FIP200 is then recruited to clustered NDP52 via its NDP52 binding site in the C-terminal part of its CC. We found that engagement to NDP52 destabilizes the conformation of the N-terminal part of the FIP200 CC and increases membrane affinity, presumably by melting the coiled coil and increasing the exposure of hydrophobic side chains in this region. These changes occur simultaneously with clustering of multiple FIP200 dimers. In our study, the role of clustering in driving membrane affinity was evidenced by the increase in binding when GST-fusions were used, because GST is dimeric. Taken together, the MR conformational change and higher-order clustering drives strong membrane recruitment.

FIP200 is thought to initially localize to phosphatidylinositol synthase (PIS)-enriched ER subdomains prior to PI(3)P generation by the autophagic PI 3-kinase complex I (*Nishimura et al., 2017*). In the presence of an ER-like synthetic lipid mixture, only the MR showed significant changes in HDX. A basic patch on the ATG13 HORMA domain structure (*Qi et al., 2015*; *Suzuki et al., 2015*) was been reported to interact with PI(3)P (*Karanasios et al., 2013*) and so sustain ULK1 complex localization at the omegasome subsequent to PI 3-kinase activation. Our synthetic lipid mixture intentionally omitted PI(3)P to reflect the earliest step in activation. The conserved EAT domain of the yeast ULK1 ortholog Atg1 binds to highly curved ER-like vesicles, but only in the absence of Atg13, and its lipid-binding role is probably important only in later steps of autophagosome biogenesis (*Lin et al., 2018*). The MR is not electropositive in character, unlike most membrane binding domains (*Hurley and Misra, 2000*), but this is in keeping with the neutral character of ER lipids. Our data suggest a model wherein NDP52 binding to the C-terminus of FIP200 CC promotes lipid binding at the MR, so allowing a tether to be formed between cargo and the ER subdomain destined to become

the omegasome. These data are consistent with the critical role of the NDP52-binding residues of FIP200 as demonstrated in both mitophagy (*Vargas et al., 2019*) and xenophagy (*Ravenhill et al., 2019*).

In this study, we focused on NDP52 rather than p62 because NDP52 binds much more tightly to FIP200, which facilitated the HDX-MS and biochemical experiments. It remains to be seen if the principles of membrane recruitment revealed here are general to p62 and other adaptors. The stimulation of the effect seen by using dimeric GST fusions of 4xUb and NDP52 show that multimerization is important for promoting membrane binding. While the binding site of NDP52 on FIP200 CC appears to be unique, clustering of cargo adaptors on ubiquitinated cargo is a general aspect of selective autophagy (*Kirkin and Rogov, 2019*; *Zaffagnini and Martens, 2016*). The role of the membrane-recruitment mechanism described here in bulk autophagy also remains to be determined, but to the extent that cargo-like ubiquitin condensates (*Zaffagnini et al., 2018*) mediate bulk autophagy, similar principles seem likely to apply.

# Materials and methods

## Key resources table

| Reagent type (species) or resource | Designation | Source or reference | Identifiers | Additional information |
|---|---|---|---|---|
| Cell line (*Homo sapiens*) | HEK GnTi | ATCC | CRL-3022 | |
| Recombinant DNA reagent | pCAG-GST-TEVcs-FIP200-MBP | This paper | | See 'plasmid construction' section. Can be obtained from the Hurley lab. |
| Recombinant DNA reagent | pCAG-Atg13 | This paper | | See 'plasmid construction' section. Can be obtained from the Hurley lab. |
| Recombinant DNA reagent | pCAG-GST-TEVcs-ATG101 | This paper | | See 'plasmid construction' section. Can be obtained from the Hurley lab. |
| Recombinant DNA reagent | pCAG-MBP-TSF-TEVcs-ULK1 | This paper | | See 'plasmid construction' section. Can be obtained from the Hurley lab. |
| Recombinant DNA reagent | pCAG-EGFP-ATG13 | This paper | | See 'plasmid construction' section. Can be obtained from the Hurley lab. |
| Recombinant DNA reagent | pCAG-GST-TEVcs-EGFP-ATG101 | This paper | | See 'plasmid construction' section. Can be obtained from the Hurley lab. |
| Recombinant DNA reagent | pCAG-GST-TEVcs-EGFP-FIP200-MBP | This paper | | See 'plasmid construction' section. Can be obtained from the Hurley lab. |
| Recombinant DNA reagent | pGST2-NDP52 | This paper | | See 'plasmid construction' section. Can be obtained from the Hurley lab. |
| Recombinant DNA reagent | pCAG-MBP-NDP52 | This paper | | See 'plasmid construction' section. Can be obtained from the Hurley lab. |
| Recombinant DNA reagent | pGEX5-4xUb | *Zaffagnini et al., 2018* | | From Sascha Martens group (Vienna) |

*Continued on next page*

*Continued*

| Reagent type (species) or resource | Designation | Source or reference | Identifiers | Additional information |
|---|---|---|---|---|
| Recombinant DNA reagent | pGST2-NDP52-mCherry | This paper | | See 'plasmid construction' section. Can be obtained from the Hurley lab. |
| Recombinant DNA reagent | pCAG-TSF-NDP52-mCherry | This paper | | See 'plasmid construction' section. Can be obtained from the Hurley lab. |
| Recombinant DNA reagent | pET-6xHis-TEVcs-NDP52 | This paper | | From Sascha Martens group (Vienna) |
| Recombinant DNA reagent | pCAG-GST-TEVcs-FIP200(1274 C) | This paper | | See 'plasmid construction' section. Can be obtained from the Hurley lab. |
| Recombinant DNA reagent | pCAG-GST-TEVcs-FIP200(N-640)-MBP | This paper | | See 'plasmid construction' section. Can be obtained from the Hurley lab. |
| Recombinant DNA reagent | pCAG-GST-TEVcs-FIP200(636-C)-MBP | This paper | | See 'plasmid construction' section. Can be obtained from the Hurley lab. |
| Recombinant DNA reagent | pCAG-GST-TEVcs-FIP200(delta790-1050, ΔMR)-MBP | This paper | | See 'plasmid construction' section. Can be obtained from the Hurley lab. |
| Recombinant DNA reagent | pCAG-GST-TEVcs-FIP200(1363-1370Mut, ΔNDP52)-MBP | This paper | | See 'plasmid construction' section. Can be obtained from the Hurley lab. |
| Recombinant DNA reagent | pCAG-GST-TEVcs-FIP200(1274 C) M1-M8 | This paper | | See 'plasmid construction' section. Can be obtained from the Hurley lab. |
| Commercial assay or kit | ADP-Glo Max Assay | Promega, Madison, WI | V6930 | |
| Software, algorithm | Proteome Discoverer 2.1 | Thermo Fisher Scientific, Waltham, MA | https://www.thermofisher.com/order/catalog/product/OPTON-30795 | |
| Software, algorithm | HDExaminer | Sierra Analytics, Modesto, CA | http://massspec.com/hdexaminer/ | |
| Software, algorithm | Nikon Elements microscope imaging software 4.60 | Nikon Corporation, Tokyo, Japan | https://www.nikoninstruments.com/Products/Software/NIS-Elements-Advanced-Research/NIS-Elements-Viewer | |
| Software, algorithm | Custom Python scripts and Jupyter notebooks | This paper | | Access at https://github.com/Hurley-Lab/FIP-NDP52-paper |
| Software, algorithm | Relion | | SCR_016274 | |
| Other | Glutathione Sepharose 4B GST-tagged protein purification resin | GE healthcare, Chicago, IL | Cat#17075605 | |
| Other | Amylose Resin | New England Biolabs, Ipswich, MA | Cat#E8021L | |

*Continued on next page*

*Continued*

| Reagent type (species) or resource | Designation | Source or reference | Identifiers | Additional information |
|---|---|---|---|---|
| Other | Strep-Tactin Superflow high capacity 50% suspension | IBA Lifesciences, Göttingen, Germany | Cat# 2-1208-010 | |

## Plasmid construction

The sequences of all DNAs encoding components of ULK1 complex were codon optimized, synthesized and then subcloned into the pCAG vector. All DNAs encoding NDP52 (except His-NDP52) were subcloned into pCAG or pGST2 vectors. The linear tetraubiqutin expressed in pGEX5 and His-NDP52 in pET were obtained from Sascha Martens (Vienna). Proteins were tagged with GST, MBP or TwinStrep-Flag (TSF) for affinity purification or pull-down assays. N-terminal GST, MBP or TSF tags were followed by a tobacco etch virus cleavage site (TEVcs). All constructs were verified by DNA sequencing. Details are shown in the key resources table.

## Protein expression and purification

GST-FIP200-MBP, FIP200(N-640)-MBP, FIP200(636-C)-MBP and full-length ULK1 complex protein samples used for NSEM, HDX-MS, fluorescent pulldown assays, ADP-Glo kinase assays and liposome sedimentation assays were expressed in HEK293-GnTI suspension cells by using the polyethylenimine (Polysciences) transfection system. Cells were transfected at a concentration of 2–3 $\times$ 10$^6$/mL and harvested after 48 hr. The harvested cells were pelleted at 500 x g for 10 min at 4℃, washed with PBS once, and then stored at −80℃. The pellets were then lysed with lysis buffer containing 50 mM Tris-HCl pH 7.4, 200 mM NaCl, 2 mM MgCl$_2$, 1 mM TCEP, 1% Triton X-100, 10% Glycerol and protease inhibitors (Roche) before being cleared at 16000 x g for 30 min at 4℃. The supernatant was then incubated with Glutathione Sepharose 4B (GE Healthcare) or Strep-Tactin Sepharose (IBA Lifesciences) as appropriate, with gentle shaking for 12 hr at 4℃. The mixture was then loaded onto a gravity flow column, and the resin was washed extensively with wash buffer (50 mM HEPES pH 8.0, 200 mM NaCl, 1 mM MgCl$_2$ and 1 mM TCEP). The proteins were eluted with wash buffer containing 50 mM glutathione or 10 mM desthiobiotin, as appropriate. To purify GST-FIP200-MBP, FIP200(N-640)-MBP, FIP200(636-C)-MBP, FIP200-MBP and EGFP-FIP200-MBP, eluted protein samples may be treated with TEV protease at 4℃ overnight before flowing through Amylose resin (New England Biolabs) for a second step of affinity purification. To purify ULK1 complex and EGFP-ULK1 complex, FIP200/ATG13/ATG101 subcomplex and ULK1 were expressed and purified separately. After the first step of affinity purification, the two samples were mixed, cleaved by TEV at 4℃ overnight, and subjected to a second step of affinity purification using the MBP tag. The eluted sample was passed through a Strep-Tactin Sepharose (IBA Lifesciences) column to clear the MBP-TSF tag from ULK1. The final buffer after MBP affinity purification is 20 mM HEPES (pH 8.0), 200 mM NaCl, 1 mM TCEP and 50 mM Maltose.

GST-NDP52, GST-NDP52-mCherry, 6xHis-NDP52 and GST-4xUb protein samples were expressed in *E. coli* (BL21DE3) at 18℃ overnight. The harvested cells were pelleted at 4500 x g for 20 min at 4℃, washed with PBS once, and then stocked in −80℃ if needed. The pellets were then suspended in a buffer containing 50 mM Tris-HCl pH 7.4, 200 mM NaCl, 2 mM MgCl$_2$, 1 mM TCEP and protease inhibitors (Roche), and sonicated before being cleared at 16000 x g for 30 min at 4℃. The supernatant was incubated with Glutathione Sepharose 4B (GE Healthcare) or Ni-NTA Resins (Qiagen) as appropriate, with gentle shaking for 2 hr at 4℃. The mixture was then loaded onto a gravity flow column, and the resin was washed extensively with wash buffer (50 mM HEPES pH 8.0, 200 mM NaCl, 1 mM MgCl$_2$ and 1 mM TCEP). The proteins were eluted with wash buffer containing 50 mM glutathione or 200 mM imidazole, as appropriate. The protein samples were applied to a final size exclusion chromatography step before use. For GST-4xUb, a Superdex 200 column (GE Healthcare) was used, and for NDP52, a Superose six column (GE Healthcare) was used. The running buffer was 20 mM HEPES (pH 8.0), 200 mM NaCl and 1 mM TCEP. For purification of TSF-NDP52-mCherry, the transfected HEK GnTI cells were harvested and treated as ULK1 transfected HEK GnTI cells mentioned above. The supernatant was incubated with Strep-Tactin Sepharose (IBA Lifesciences) with gentle

shaking for 2 hr at 4°C, and then subjected to a Strep tag affinity purification mentioned above. The protein samples were applied to a final size exclusion chromatography step using a Superose six column (GE Healthcare).

## Negative stain electron microscopy collection and coiled-coil tracing

Grids were glow discharged in PELCO easiGlow for 25 s at 25 mAmps. Full length FIP200 labeled with an N terminal GST tag and a C terminal MBP tag was incubated on continuous carbon grids at 100 nM concentration. Protein was stained with 2% uranyl formate twice before drying and imaging. Samples were imaged with a T12 operating at 120 kV at a nominal magnification of 49,000x. This corresponds to 2.2 Å/pixel on a Gatan CCD 4k × 4 k camera. Data were collected at 60 e-/Å$^2$ and single particles were manually selected within Relion. Particles were binned and extracted in a box size of 120 by 120 with 8.8 Å/pixel.

2D classification of the N and C terminal domains was carried out as follows. Relion LoG picker was used to pick the negative stain micrographs. 17,281 single particles were classified into 200 classes. Resolved class averages were selected based on the shape of the NTD dimer and the CTD trimeric density (the dimer Claw domain and 2 MBP tags). A second round of classification was performed to better resolve the class averages (*Figure 1—figure supplement 1B–C*).

Full length single particles were imported into FIJI ImageJ and traced using the plugin Simple Neurite Tracer yielding 117 individual tracks. Tracks were started at the beginning of the coiled-coil domain after the distinct NTD crescent shape and ended between the double MBP density at the C terminus. Tracking was checked for each single particle to ensure accuracy of the automatically determined path. End to end distances and total length measurements are shown as histograms.

## Hydrogen-deuterium exchange mass spectrometry

FIP200 and NDP52 samples for HDX were concentrated to a 20 µM stock solution, ULK1 complex was concentrated to a 10 µM stock solution, while POPS/POPI SUVs (molar ratio = 1:1) were prepared as a 1 mM stock solution. The buffer used was 20 mM HEPES (pH 8.0), 200 mM NaCl and 1 mM TCEP. To prepare the ULK1c/NDP52 sample or ULK1c/SUV sample, 2 µL of ULK1c stock solution was mixed with 8 µL of NDP52 or SUV stock solution and then incubated at 23°C for 30 min. To prepare the FIP200 or ULK1c sample, 2 µL of protein stock was mixed with 8 µL of control buffer (20 mM HEPES (pH 8.0), 200 mM NaCl, 1 mM TCEP), and then incubated at 23°C for 30 min. Exchange was initiated by adding 90 µL of D$_2$O buffer containing 20 mM HEPES (pH 8.0), 200 mM NaCl, 1 mM TCEP into 10 µL of protein mixture at 30°C. Exchange was carried out for 6 s, 60 s, 600 s, or 60000 s, and quenched at 0°C by the addition 100 µL of ice-cold quench buffer (400 mM KH$_2$PO$_4$/H$_3$PO$_4$, pH 2.2). The 60000 s sample served as the maximally labeled control. All HDX reactions were repeated three times. Quenched samples were injected into a chilled HPLC (Agilent) setup with in-line peptic digestion and desalting steps. The analytical column used was a Biobasic 8.5 µm KAPPA column (Fisher Scientific). The peptides were eluted with an acetonitrile gradient and electrosprayed into an Orbitrap Discovery mass spectrometer (Thermo Scientific) for analysis. To generate the gradient, solvent A was 0.05% TFA, while solvent B was 0.05% TFA in 90% acetonitrile. The elution method was as follows: 0–6 min: 10% B; 6–42 min: from 10% B to 55% B; 42–43 min: from 55% B to 90% B; 43–53 min: 90% B; 53–54 min: from 90% B to 10% B; 54–60 min: 10% B. The spray voltage was set at 3.4 kV, capillary temperature was set at 275°C, capillary voltage was set at 37 V and tube-lens was set at 120 V. As a control, unexchanged samples went through the same process, except that D$_2$O was replaced by H$_2$O.

To identify peptides, unexchanged samples were analyzed by tandem MS/MS analysis with the same HPLC method. Tandem MS/MS was performed using data dependent analysis, in which a cycle of one full-scan MS spectrum (m/z 200–2000) was acquired followed by MS/MS events (CID fragmentation). MS/MS was sequentially generated on the ten most intense ions selected from the full MS spectrum at a 35% normalized collision energy. The ion trap analyzer was used for MS2, activation time was 30 ms, and the dynamic exclusion was set at 30 s. For HDX mass analysis, only a full-scan MS spectrum was acquired, and the resolution was 30000. Database searches were performed with Proteome Discoverer 2.1 (Thermo Fisher Scientific) using the Sequest HT search engine to identify peptides. Raw data were searched against the small database containing all four components of ULK1 complex. The following search parameters were used: unspecific cleavage was used; precursor

mass tolerance was set to ±10 ppm and fragment mass tolerance was set to ±0.6 Da. Target FDR was set to 1% as the filter cut-off for the identified peptides. For HDX analysis, mass analysis of the peptide centroids was performed using HDExaminer (Sierra Analytics), followed by manual verification for every peptide. The HDX-MS data are provided in *Supplementary file 2*.

## ADP-Glo kinase assay

100 nM purified ULK1 complex was mixed with 5 µM ULKtide (SignalChem Biotech Inc), and incubated at room temperature for 1 hr. The reaction buffer was 20 mM HEPES pH 8.0, 200 mM NaCl, 2 mM MgCl$_2$, 100 µM ATP, 20 mM Maltose and 1 mM TCEP. The reaction was terminated by adding an ATP-depletion reagent. Then a kinase detection reagent was added to convert ADP to ATP, which is used in a coupled luciferase reaction. The luminescent output was measured with a GloMax-Multi detection system (Promega) and was correlated with the kinase activity.

## Pull-down assays

For the fluorescence-based pulldown assay in *Figure 1—figure supplement 2D*, purified fluorescent ULK1 complex (FIP200-MBP/EGFP-ATG13/EGFP-ATG101/ULK1) was mixed with purified GST-NDP52 or GST-4xUb and 10 µL Glutathione Sepharose 4B (GE Healthcare). The final buffer was 20 mM HEPES pH 8.0, 200 mM NaCl, 1 mM TCEP, 10 mM maltose and 1% Triton-X-100. The final protein concentration was 50 nM ULK1 complex and 1 µM GST-NDP52 or GST-4xUb. The final volume was 150 µL. The system was gently shaken for 1 hr at 23℃. The EGFP signal of the supernatant was measured before and after shaking to calculate the pull-down efficiency.

For the pulldown assay in *Figure 2E*, purified fluorescent FIP200(EGFP-FIP200-MBP) or ULK1 complex (EGFP-FIP200-MBP/ATG13/ATG101/ULK1) was mixed with purified GST-NDP52 at the indicated concentrations.

For the pulldown assay in *Figures 2F* and 10 mL of HEK293-GnT1 suspension cells were transfected at the concentration of 2–2.5 × 10⁶/mL and harvested after 48 hr. The harvested cells were pelleted at 500 x g for 8 min at 4℃, and then washed with 5 mL PBS once. The pellets were then lysed with 1 mL lysis buffer containing 50 mM Tris-HCl pH 7.4, 200 mM NaCl, 2 mM MgCl$_2$, 1 mM TCEP, 1% Triton X-100, 10% Glycerol and protease inhibitors (Roche) before being cleared at 12000 rpm for 10 min at 4℃. The supernatant was then incubated with 20 µL Glutathione Sepharose 4B (GE Healthcare) or Amylose resin (New England Biolabs) with gentle shaking for 8 hr at 4℃. The protein-bound resin was washed with 1 mL lysis buffer three times, and then eluted with 60 µl elution buffer containing 50 mM glutathione or 50 mM maltose, respectively. The eluted proteins were applied to SDS–PAGE for analysis.

## Liposome sedimentation assays

POPI, POPS, POPE and POPC from Avanti Polar Lipids, Inc were dissolved in chloroform. The chloroform was removed by overnight incubation under vacuum. Lipids were rehydrated in 20 mM HEPES pH 8.0, 200 mM NaCl, 1 mM TCEP and 50 mM Maltose for 30 min on ice and then resuspended by vigorous vortexing. Small unilamellar vesicles (SUVs) were prepared by sonication on ice until the solution appeared clear. Two kinds of SUV were prepared. PS/PI SUV is comprised of 50% POPS and 50% POPI, while PC/PE SUV is comprised of 50% POPC and 50% POPE. The protein and SUV were mixed and incubated at room temperature for 30 min. The final concentration of protein is 1 µM and the final concentration of liposomes was 100 µM. The final buffer was 20 mM HEPES pH 8.0, 200 mM NaCl, 1 mM TCEP and 10 mM Maltose. The liposomes were pelleted in an ultracentrifuge (TLA100 rotor) for 45 min at 50,000 rpm, and the presence of protein in the pellet and supernatant fractions was analyzed by SDS-PAGE.

## Preparation of giant unilamellar vesicles (GUVs)

GUVs were prepared by hydrogel-assisted swelling as described previously (*Fracchiolla et al., 2020*). Polyvinyl alcohol (PVA) with a molecular weight of 145,000 (Millipore) was used as hydrogel substrate. 300 µL of 5% (w/w) PVA solution was spin coated onto a plasma-cleaned coverslip of 25 mm diameter. The coated coverslip was placed for 30 min in a heating incubator at 60℃ to dry the PVA film. A lipid mixture with a molar composition of 64.8% DOPC, 20% DOPE, 10% POPI, 5% DOPS and 0.2% Atto647N DOPE at 1 mg/ml was spread uniformly onto the PVA film. The lipid-

coated coverslip was then put under vacuum overnight to evaporate the solvent. 300 µL 400 mOsm sucrose solution was used for swelling for 1 hr at room temperature, and the vesicles were then harvested and used immediately.

## Membrane protein recruitment – GUV assay

The reactions were set up in an eight-well observation chamber (Lab Tek) at room temperature. The chamber was coated with 5 mg/ml β casein for 30 min and washed three times with reaction buffer (20 mM HEPES at pH 8.0, 190 mM NaCl and 1 mM TCEP). A final concentration of 10 µM GST-4xUb, 1 µM NDP52, and 100 nM GFP-FIP200 or GFP-ULK1 complex was used for all reactions unless otherwise specified. 15–20 µL GUVs were added to initiate the reaction in a final volume of 150 µL. After 5 min incubation, during which we picked random views for imaging, time-lapse images were acquired in multitracking mode on a Nikon A1 confocal microscope with a 63 × Plan Apochromat 1.4 NA objective. Three biological replicates were performed for each experimental condition. Identical laser power and gain settings were used during the course of all conditions.

## Microscopy-based bead protein-protein interaction assay

A mixture of 1 µM GST-NDP52 and 100 nM GFP-ULK1 complex was incubated with 10 µL Glutathione Sepharose beads (GE Healthcare) in a reaction buffer containing 20 mM HEPES at pH 8.0, 200 mM NaCl and 1 mM TCEP. After incubation at room temperature for 30 min, the beads were washed three times, suspended in 120 µL reaction buffer, and then transferred to the observation chamber for imaging. Images were acquired on a Nikon A1 confocal microscope with a 63 × Plan Apochromat 1.4 NA objective. Three biological replicates were performed for each experimental condition.

## Image quantification

GUV images were analyzed using a custom script implemented in Python 3.6 (https://github.com/Hurley-Lab/FIP-NDP52-paper; *Jensen, 2020*; copy archived at https://github.com/elifesciences-publications/FIP-NDP52-paper). First, to obtain the outline of all the vesicles within a field of view, images were segmented into regions corresponding to local maxima of the membrane fluorescence channel, which were defined by applying an Otsu threshold to the differences between local maxima and minima. Then, binding of fluorescently labeled proteins was quantified by taking the mean value of these segmented pixels in the fluorescent protein channel. Background was calculated as the average of the vesicle-internal background and the vesicle-external background and subtracted from the fluorescence signal. The intensity trajectories of multiple fields of view were then obtained frame by frame. Multiple intensity trajectories were calculated, and the average and standard deviation calculated and reported.

For quantification of bead binding, the outline of individual beads was manually defined based on the bright field channel. The intensity threshold was calculated by the average intensities of pixels inside and outside of the bead and then intensity measurements of individual bead were obtained. Averages and standard deviations were calculated among the measured values per each condition and plotted in a bar graph.

## Acknowledgements

We thank P Grob for EM facility support. S Martens and D Fracchiolla for discussions and for the His-NDP52 and GST-4xUb constructs, and S von Bülow for discussions. This study was supported by the National Institutes of Health R01 GM111730 and Human Frontier Science Program RGP0026/2017 to JHH, a Jane Coffin Childs Memorial Fund Fellowship to A L Y, and a Tang Visiting Scholar fellowship to CC.

## Additional information

### Competing interests

James H Hurley: JHH is co-founder of Casma Therapeutics. The other authors declare that no competing interests exist.

### Funding

| Funder | Grant reference number | Author |
|---|---|---|
| National Institutes of Health | R01 GM111730 | James H Hurley |
| Human Frontier Science Program | RGP0026/2017 | James H Hurley |
| Jane Coffin Childs Memorial Fund for Medical Research | | Adam L Yokom |
| University of California Berkeley | Tang Distinguished Scholarship | Chunmei Chang |

The funders had no role in study design, data collection and interpretation, or the decision to submit the work for publication.

### Author contributions

Xiaoshan Shi, Chunmei Chang, Conceptualization, Data curation, Formal analysis, Validation, Investigation, Visualization, Writing - original draft; Adam L Yokom, Conceptualization, Data curation, Formal analysis, Funding acquisition, Validation, Investigation, Visualization, Writing - original draft; Liv E Jensen, Software, Formal analysis; James H Hurley, Conceptualization, Supervision, Funding acquisition, Writing - original draft, Project administration

### Author ORCIDs

Xiaoshan Shi (iD) https://orcid.org/0000-0001-7931-8684
Chunmei Chang (iD) https://orcid.org/0000-0002-5607-7985
Adam L Yokom (iD) https://orcid.org/0000-0002-3746-7961
James H Hurley (iD) https://orcid.org/0000-0001-5054-5445

### Decision letter and Author response

Decision letter https://doi.org/10.7554/eLife.59099.sa1
Author response https://doi.org/10.7554/eLife.59099.sa2

## Additional files

### Supplementary files

• Supplementary file 1. Table S1. Statistics of HDX differences for *Figure 2D* The peptides covering residue 800–1250 of FIP200 are listed, and the number of repeat and the sequence of each peptides are provided. The statistical test of HDX differences employed is paired T-test. P value less than 0.05 is highlighted in red, representing significant difference between ULK1 complex and ULK1 complex with NDP52 samples.

• Supplementary file 2. HDX-MS Data Sets The summary tables of HDX data for each protein in ULK1 complex are provided, and the original peptide pool results are included.

• Transparent reporting form

### Data availability

All data generated in this study are included in the manuscript and supporting files. HDX-MS data are included as Supplementary Data Set 1.

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
