## [Decision Letter]

**Acceptance summary:**

This paper employs biophysical approaches to examine membrane binding of NDP52, a protein involved in selective autophagy. The authors propose a new membrane binding mechanism involving the coiled-coil of NDP52. This represents a new mechanism that may collaborate with ubiquitin binding by NDP52 to capture autophagic cargo.

**Decision letter after peer review:**

Thank you for submitting your article "The autophagy adaptor NDP52 and the FIP200 coiled-coil allosterically activate ULK1 complex membrane recruitment" for consideration by *eLife*. Your article has been reviewed by three peer reviewers, including Wade Harper as the Reviewing Editor and Reviewer #1, and the evaluation has been overseen by Suzanne Pfeffer as the Senior Editor.

The reviewers have discussed the reviews with one another and the Reviewing Editor has drafted this decision to help you prepare a revised submission.

Summary:

This paper describes biophysical and structural studies on purified, full length FIP200 in complex with its other binding partners, including NDP52 (a well-known ubiquitin binding autophagy receptor). The results presented in this manuscript corroborate prior results from the Hurley lab demonstrating a "C-Shaped" N-terminal dimer that acts as a scaffold of ULK1-ATG13-ATG101, while the C-terminal claw region engages receptor molecules such as NDP52 and p62. Furthermore, Shi et al. conclude that a coiled-coil region within FIP200 undergoes a dynamic conformational change upon binding with the receptor molecule NDP52, increasing its affinity for FIP200 towards membranes. They propose that NDP52 binding to the surface to cargo elicits a conformational change in the FIP200, increasing its propensity to bind to membranes and drives the local clustering of the ULK1 kinase leading to its autoactivation and the induction of autophagy.

While the reviewers feel that the work is of high quality and is potentially appropriate for publication in *eLife*, the consensus after extensive discussion among the reviewers and the senior editor is that core elements of the proposed model are not fully supported by the data presented thus far. This has to do primarily with the relative roles/contributions of coiled-coli domains in FIP200 and NDP52 for association with membranes, and whether there is a key role for allostery in the mechanism of membrane binding. Another weakness in the paper is that the work primarily employs liposomes, rather than intact membranes, and there isn't data supporting the role for these structural elements in autophagy within cells. Given this, we are offering the possibility of a significant revision, but without the normal time constraints, due to COVID-19 related limitations on lab access.

Essential revisions:

The reviewers would like to see:

1) A better understanding of the structural features within the coiled-coils that facilitate membrane binding.

2) An analysis of the interactions with biological membranes rather than liposomes would be important in further validating the model.

3) A better understanding of the allostery mechanism is needed. It is often assumed that regions predicted to be "coiled-coil" form a typical helical dimeric assembly of α-helices. In such case, the "hydrophobic residues", which are assumed to be buried in the dimeric interface, twists along the long axis of the coiled-coil. The MR reported in this study is fairly long (over 150 residues). Although association of short coiled-coil domains are known to associate with membranes, it is harder to see how an extended helix might do this, and so the question is whether this coiled-coil undergoes an alteration in structure (e.g. melting) that is in some way coupled with the allostery (or not). If so, that could cause liposome binding, as shown in Figure 3D. Again, Figure 3D's data are without NDP52. As such, the data does not directly demonstrate the suggested binding mode (hydrophobic residues are exposed upon allosteric conformational change, and those residues bind to membranes).

After discussion, we concluded that perhaps analysis of mutants in cells is beyond the scope of this heavily biophysical analysis. However, one aspect that is unclear concerns the conclusion that NDP52 is effectively recruited to membranes on its own or together with FIP200. For example, in the context of mitophagy, mutations in the ubiquitin binding domain of NDP52 abolish recruitment to the mitochondrial outermembrane downstream of parkin activation in cells. It is difficult to see how the intrinsic membrane binding of NDP52 could be operative in this context as the core mechanism of recruitment, given the ubiquitin binding dependence. Although OPTN is the main adaptor in this form of selective autophagy, NDP52 plays a partially redundant role and cells only expressing NDP52 as the ubiquitin binding autophagy adaptor can support mitophagy. So this aspect of the model for direct and biologically relevant membrane as the primary mechanism for membrane recruitment doesn't precisely fit with prior data and it may be necessary to discuss how membrane binding may be relevant in the context of other mechanisms that place NDP52 adjacent to the membrane. Might the primary recruitment mechanism be through ubiquitin, but then the membrane interactions serve to orient the molecules on the membrane surface for optimal downstream functions?

Reviewer #1:

Autophagy is driven by the action of the central and highly conserved ULK1 complex which comprises of the Serine/Threonine kinase ULK1, the large scaffolding protein FIP200, and the accessory proteins ATG13 and ATG101. Additionally, cargo sequestration in autophagy is bought about by the binding of autophagy "receptor" molecules to the surface of cargo and helps recruit the ULK1 complex via direct interactions to promote their degradation.

Structural studies on this complex (specifically FIP200) have been limited, largely in part due the size and the prevalence of an Intrinsically disordered region in the center of the protein. Recently, the Hurley lab have made strides in the understanding of three-dimensional architecture of this complex by ascertaining the structure of the N-terminal(NTD) and extreme C-terminal regions(Claw) of FIP200. In this paper, Shi and colleagues describe another advancement in the study of this complex by biophysical and structural studies on purified, full length FIP200 in complex with its other binding partners. The results presented in this manuscript corroborate prior results from the Hurley lab demonstrating a "C-Shaped" N-terminal dimer that acts as a scaffold of ULK1-ATG13-ATG101, while the C-terminal claw region engages receptor molecules such as NDP52 and p62. Furthermore, Shi et al. demonstrate that a coiled-coil region within FIP200 undergoes a dynamic conformational change upon binding with the receptor molecule NDP52, increasing its affinity for FIP200 towards membranes. Therefore, autophagy receptor binding to the surface to cargo elicits a conformational change in the FIP200, increasing its propensity to bind to membranes and drives the local clustering of the ULK1 kinase leading to its autoactivation and the induction of autophagy.

Overall this is a well-executed and presented study and would be suitable for publication once the following points are addressed.

1) While the biophysical assays are relatively convincing, one obvious drawback is the absence of any functional autophagy assays to monitor if this receptor-mediated conformational change is actually physiologically important. The most obvious experiment in this regard would be to add back the FIP200 (deleted membrane region) and monitor any form of autophagy. However, given the state of affairs due to the ongoing COVID19 pandemic I would say that this experiment is not critical, but if performed would significantly bolster the conclusions of this paper.

2) The HDX graphs were not always clear to this reader. For example, in Figure 1C the blue and grey colors need to be assigned to the relevant sample. Additionally, the number of repeats performed here and in every experiment presented should be stated.

3) The increase in the FIP200 residues 800-1250 (2D) after NDP52 doesn't seem significant by the authors own cut-offs (> or <10%), nevertheless this region appears to the important for membrane binding.

4) Figure 2F is lacking input controls to show that the levels of MBP-NDP52 and GST FIP200 in cells are equal.

Reviewer #2:

In this manuscript from Hurley and colleagues, the authors describe for the first time the structure of the full-length FIP200 protein including its ~75 nm long coiled-coil domain, and then explore how this region of the protein contributes to the membrane targeting of FIP200 and of its cognate kinase, ULK1. Their principle discoveries are that the coiled coil includes a region that can interact with acidic lipids PI and/or PS, that this motif can adopt a range of structures (fully elongated or bent back on itself) and that there is no allosteric communication between the opposite termini of the coiled-coil, thus for example cargo-binding does not directly alter ULK1 interactions. Furthermore, the membrane interaction is augmented under conditions of multimerization, such as occurs when FIP200 is engaged with the oligomeric cargo adaptor NDP52 and polyUB.

The strength of the paper is the excellent in vitro biochemistry and biophysics. The authors use deuterium exchange to establish regions on FIP200 involved in ULK1, NDP52 and membrane binding. They use electron microscopy to develop a model of the full-length protein (and protein complex). GUV recruitment assays confirm that membrane binding of ULK1 depends upon FIP200 and NDP52/Ub, and that this membrane binding is augmented by the motif in the coiled-coil domain. Further, membrane binding leads to localized domains of ULK1 protein, consistent with the association of ULK1 to an oligomerizing cargo/cargo-adaptor complex. The design and chronology of experiments are logical and each is well controlled.

1) I have only one relatively significant concern with the structure of the narrative. The authors pose the general question of how FIP200 organizes ULK1 on membranes and attempt to address this question through most of the paper without acknowledging that NDP52 itself can and does bind membranes. This point, previously published, is finally addressed at the end of the Results, but it fundamentally changes the way the results should be considered. In the Discussion they say " The data presented here suggest that the membrane binding region (MR) in the N terminal portion of FIP200 CC is the major membrane binding site responsible for the first ULK1 complex membrane recruitment event in autophagy " and "Our data suggest a model wherein NDP52 binding to the Cterminus of FIP200 CC promotes lipid binding at the MR, so allowing a tether to be formed between cargo and the ER subdomain destined to become the omegasome." However, their GUV recruitment assay (Figure 5 and Figure 5—figure supplement 1) strongly implies that although the membrane-binding motif they have described on FIP200 can augment membrane binding, NDP52 is still largely sufficient and a/the principle driver of membrane targeting. This element of the model should be more fully described and the existing literature on the role of NDP52 in targeting these complexes specifically to membranes should be introduced early.

Reviewer #3:

The authors dissect protein-protein interactions within the NDP52-ULK1 complex and its membrane interaction using HDX and identify a membrane-interacting region in the coiled-coil domain of its component FIP200. They show the ULK1 complex is recruited to GUV membranes through the identified membrane-binding region, and this membrane recruitment is activated by NDP52. Because NDP52 binds to another region in the coiled-coil domain of FIP200, they conclude that NDP52 allosterically activates the membrane-binding region of the FIP200. This in vitro study is performed carefully, and the presented data appear to be of high quality. Identification of the membrane-interacting region of FIP200 is valuable for the autophagy field and would help future work toward the elucidation of the role of FIP200 in autophagy. However, it seems that the allosteric activation is not fully supported by the data and the work struggles to establish the biological significance of this membrane-binding activity by FIP200.

1) It is not clear how the FIP200-mediated membrane association fits in the biological event (selective autophagy). There seem to be several candidate membranes that FIP200 might associate with: the membrane of the targets (bacteria/mitochondria), the ER, the omegasome, the autophagosome/phagophore. However, which of these is the bona fide membrane target of FIP200 is not clear, making it difficult to relate the findings of this work to the mechanism of selective autophagy. NDP52 is also shown to be capable of associating with membranes, which complicates the interpretation. How are the roles of these membrane associations by FIP200 and NDP52 related?

2) The coiled-coil domain is suggested to associate with membranes through hydrophobic residues exposed upon NDP52 binding. This is a very interesting and thoughtful model deduced from a careful interpretation of the HDX data. However, as hydrophobic residues could bind to many things, this explanation alone is probably not sufficient to convince that the observed membrane-binding is biological. More detailed mechanism of the membrane association could help clarify this. Is the MR predicted to form a typical parallel coiled-coil dimer? If so, would a structural model provide insights into the mechanism of membrane binding?

3) The authors suggest that NDP52 allosterically activates the membrane binding by FIP200. However, FIP200 is suggested to have weak and intrinsic membrane-binding activity (in the absence of NDP52) (Figure 3D). The overall membrane binding could be enhanced through multivalent FIP200-membrane interactions created by GST-4xUb-mediated clustering. Thus, it seems that the membrane-binding data of the NDP52-ULK1 complex can be explained without invoking such allosteric activation. The NDP52-membrane interactions would also become multivalent in the presence of GST-4xUb so that they would also contribute to the membrane recruitment of the overall complex. Thus, with two membrane-binding domains and clustering, it seems to be not straightforward to determine the magnitude of allosteric activation in the presented data.

4) NDP52 alone does not seem to recruit FIP200 to membranes. However, NDP52 alone is shown to be capable of destabilizing the MR. GST-NDP52 enhances the FIP200 membrane association but only modestly. Thus, these data seem to suggest that the membrane-binding affinity of "activated MR" is very low. If hydrophobic residues in such a long region of the coiled-coil were exposed to interact with membranes, as suggested, shouldn't the affinity be higher?

5) Direct evidence for the allosteric activation is lacking. While structural information is limited to at a low-resolution, the membrane-binding region and NDP52 binding site are both in the coiled-coil. Thus, it may be possible to design and create a mutant FIP200 that can bind to NDB52 but not to membranes without touching MR and NDP52 binding sites directly. Such a mutant protein will be strong support for the suggested allostery.

6) Is the mechanism by which NDP52 associates with membranes known? The authors should create an NDP52 construct that can bind to FIP200 but not to membranes and use it in their ULK1/FIP200 membrane recruitment assays to remove the contribution of NDP52-membrane association.

7) Full-length FIP200 should be included in the sedimentation assay in Figure 3D as it would clarify whether the full-length and the truncated proteins are different in membrane binding.

8) An MR-only construct should be tested for membrane binding using sedimentation assays and GUV assays to clarify whether the activity of the MR is intrinsically suppressed or not. The MR is said to be not electropositive, but in Figure 3, the C-terminal FIP200 construct shows a better binding with negatively charged liposomes. An MR-only construct might reveal the differences between the MR and the entire CTD. Is the CTD overall electropositive?

9) The oligomeric state of NDP52 should be examined as it has a coiled-coil domain that may oligomerize.

10) The KD between NDP52 and FIP200 coiled-coil would be helpful for understanding the contribution of FIP200 in membrane association if available.

11) It may be helpful to present the raw HDX protection profiles (not differences between two samples) as supplements to show protection levels of each samples. Are the coiled-coil regions well protected in the absence of NDP52/liposomes?

---

## [Author Response]

Essential revisions:The reviewers would like to see:1) A better understanding of the structural features within the coiled-coils that facilitate membrane binding.

The HDX data are consistent with melting of the coiled coil. Our model is that when bound to the membrane, these regions are not coiled coils at all, but rather are unfolded. We are not sure how the reviewers would propose that we should determine the structural feature of unfolded regions protruding into the membrane, beyond the HDX we already provided. These are too disordered for crystallography and too small and dynamic for cryo-EM. HDX is ideally suited to exploring this type of system, and indeed we consider it to be the gold standard for this type of question. This is the reason we focused the paper on these data. We have added a new Figure 6 to be more clear about what we think is happening to the coiled coils.

2) An analysis of the interactions with biological membranes rather than liposomes would be important in further validating the model.

We appreciate and concur with the comment below that "analysis of mutants in cells is beyond the scope" of our study.

3) A better understanding of the allostery mechanism is needed. It is often assumed that regions predicted to be "coiled-coil" form a typical helical dimeric assembly of α-helices. In such case, the "hydrophobic residues", which are assumed to be buried in the dimeric interface, twists along the long axis of the coiled-coil. The MR reported in this study is fairly long (over 150 residues). Although association of short coiled-coil domains are known to associate with membranes, it is harder to see how an extended helix might do this, and so the question is whether this coiled-coil undergoes an alteration in structure (e.g. melting) that is in some way coupled with the allostery (or not).

Yes, the HDX data are most consistent with melting. We have made this more explicit in the new Figure 6.

If so, that could cause liposome binding, as shown in Figure 3D. Again, Figure 3D's data are without NDP52. As such, the data does not directly demonstrate the suggested binding mode (hydrophobic residues are exposed upon allosteric conformational change, and those residues bind to membranes).

The liposome sedimentation in Figure 3D was intended as a gross mapping of the membrane binding to the C-terminal as opposed to N-terminal data. Our conclusions about allostery build on this and rely principally on the HDX data and GUV reconstitutions. Certainly these conclusions could not have been drawn solely based on the data in Figure 3D.

After discussion, we concluded that perhaps analysis of mutants in cells is beyond the scope of this heavily biophysical analysis. However, one aspect that is unclear concerns the conclusion that NDP52 is effectively recruited to membranes on its own or together with FIP200. For example, in the context of mitophagy, mutations in the ubiquitin binding domain of NDP52 abolish recruitment to the mitochondrial outermembrane downstream of parkin activation in cells. It is difficult to see how the intrinsic membrane binding of NDP52 could be operative in this context as the core mechanism of recruitment, given the ubiquitin binding dependence. Although OPTN is the main adaptor in this form of selective autophagy, NDP52 plays a partially redundant role and cells only expressing NDP52 as the ubiquitin binding autophagy adaptor can support mitophagy. So this aspect of the model for direct and biologically relevant membrane as the primary mechanism for membrane recruitment doesn't precisely fit with prior data and it may be necessary to discuss how membrane binding may be relevant in the context of other mechanisms that place NDP52 adjacent to the membrane. Might the primary recruitment mechanism be through ubiquitin, but then the membrane interactions serve to orient the molecules on the membrane surface for optimal downstream functions?

Hopefully the new Figure 6 and associated discussion explains how we think our results fit into the context of the known linkage between Ub, NDP52, and FIP200. See further discussion below in the responses to individual reviewers.

Reviewer #1:Autophagy is driven by the action of the central and highly conserved ULK1 complex which comprises of the Serine/Threonine kinase ULK1, the large scaffolding protein FIP200, and the accessory proteins ATG13 and ATG101. Additionally, cargo sequestration in autophagy is bought about by the binding of autophagy "receptor" molecules to the surface of cargo and helps recruit the ULK1 complex via direct interactions to promote their degradation.Structural studies on this complex (specifically FIP200) have been limited, largely in part due the size and the prevalence of an Intrinsically disordered region in the center of the protein. Recently, the Hurley lab have made strides in the understanding of three-dimensional architecture of this complex by ascertaining the structure of the N-terminal(NTD) and extreme C-terminal regions(Claw) of FIP200. In this paper, Shi and colleagues describe another advancement in the study of this complex by biophysical and structural studies on purified, full length FIP200 in complex with its other binding partners. The results presented in this manuscript corroborate prior results from the Hurley lab demonstrating a "C-Shaped" N-terminal dimer that acts as a scaffold of ULK1-ATG13-ATG101, while the C-terminal claw region engages receptor molecules such as NDP52 and p62. Furthermore, Shi et al. demonstrate that a coiled-coil region within FIP200 undergoes a dynamic conformational change upon binding with the receptor molecule NDP52, increasing its affinity for FIP200 towards membranes. Therefore, autophagy receptor binding to the surface to cargo elicits a conformational change in the FIP200, increasing its propensity to bind to membranes and drives the local clustering of the ULK1 kinase leading to its autoactivation and the induction of autophagy.Overall this is a well-executed and presented study and would be suitable for publication once the following points are addressed.1) While the biophysical assays are relatively convincing, one obvious drawback is the absence of any functional autophagy assays to monitor if this receptor-mediated conformational change is actually physiologically important. The most obvious experiment in this regard would be to add back the FIP200 (deleted membrane region) and monitor any form of autophagy. However, given the state of affairs due to the ongoing COVID19 pandemic I would say that this experiment is not critical, but if performed would significantly bolster the conclusions of this paper.

Thanks for your understanding.

2) The HDX graphs were not always clear to this reader. For example, in Figure 1C the blue and grey colors need to be assigned to the relevant sample. Additionally, the number of repeats performed here and in every experiment presented should be stated.

Thanks for the suggestion. We have added this information to the figure legend.

3) The increase in the FIP200 residues 800-1250 (2D) after NDP52 doesn't seem significant by the authors own cut-offs (> or <10%), nevertheless this region appears to the important for membrane binding.

The 10% cut-off was arbitrary and reflected a cautious figure well in excess of the actual threshold of significance. For the region in question, we have added a table of p-values, which is the new Table S2. We have not done this throughout as the other effects, such as NDP52 binding, are so large and unambiguous.

4) Figure 2F is lacking input controls to show that the levels of MBP-NDP52 and GST FIP200 in cells are equal.

In this experiment, MBP-NDP52 and GST-FIP200(1274-C) were co-expressed in HEK cells. The cell lysate was divided into two fractions. One for MBP pulldown and the other for GST pulldown assay. Therefore, the MBP-NDP52 bands in the MBP pulldown and GST-FIP200 intensity in the GST serve as input controls, and do show equal amounts of MBP-NDP52 and GST-FIP200 in all samples.

Reviewer #2:In this manuscript from Hurley and colleagues, the authors describe for the first time the structure of the full-length FIP200 protein including its ~75 nm long coiled-coil domain, and then explore how this region of the protein contributes to the membrane targeting of FIP200 and of its cognate kinase, ULK1. Their principle discoveries are that the coiled coil includes a region that can interact with acidic lipids PI and/or PS, that this motif can adopt a range of structures (fully elongated or bent back on itself) and that there is no allosteric communication between the opposite termini of the coiled-coil, thus for example cargo-binding does not directly alter ULK1 interactions. Furthermore, the membrane interaction is augmented under conditions of multimerization, such as occurs when FIP200 is engaged with the oligomeric cargo adaptor NDP52 and polyUB.The strength of the paper is the excellent in vitro biochemistry and biophysics. The authors use deuterium exchange to establish regions on FIP200 involved in ULK1, NDP52 and membrane binding. They use electron microscopy to develop a model of the full-length protein (and protein complex). GUV recruitment assays confirm that membrane binding of ULK1 depends upon FIP200 and NDP52/Ub, and that this membrane binding is augmented by the motif in the coiled-coil domain. Further, membrane binding leads to localized domains of ULK1 protein, consistent with the association of ULK1 to an oligomerizing cargo/cargo-adaptor complex. The design and chronology of experiments are logical and each is well controlled.1) I have only one relatively significant concern with the structure of the narrative. The authors pose the general question of how FIP200 organizes ULK1 on membranes and attempt to address this question through most of the paper without acknowledging that NDP52 itself can and does bind membranes. This point, previously published, is finally addressed at the end of the Results, but it fundamentally changes the way the results should be considered. In the Discussion they say " The data presented here suggest that the membrane binding region (MR) in the N terminal portion of FIP200 CC is the major membrane binding site responsible for the first ULK1 complex membrane recruitment event in autophagy " and "Our data suggest a model wherein NDP52 binding to the Cterminus of FIP200 CC promotes lipid binding at the MR, so allowing a tether to be formed between cargo and the ER subdomain destined to become the omegasome." However, their GUV recruitment assay (Figure 5 and Figure 5—figure supplement 1) strongly implies that although the membrane-binding motif they have described on FIP200 can augment membrane binding, NDP52 is still largely sufficient and a/the principle driver of membrane targeting. This element of the model should be more fully described and the existing literature on the role of NDP52 in targeting these complexes specifically to membranes should be introduced early.

The membrane binding by the MR mutant is reduced by about 2.5-fold. Put in perspective, the MR accounts for more than half of the bound proportion. Given that collectively FIP200 and NDP52 comprise an assembly of over 2000 amino acids, it is not surprising that there are some additional contributions. We believe the residual membrane binding is due to a combination of smaller contributions from membrane binding sites on other parts of FIP200 and on NDP52. There is only one paper that we are aware of that previously proposed NDP52 bound to membranes, that of H. Yu et al. (2019). This paper concerned microtubule regulation in mitosis, not autophagy. It was found that NDP52 only bound to synthetic liposomes containing phosphatidic acid (PA) in that study. A putative PA sequence was reported in this study. We have not been able to reproduce these results in the GUV system. Using PA-containing GUVs, we observed a limited degree of binding by wild-type NDP52 or the putative PA-binding mutant (Author response 1). Given the uncertain relationship of the putative PA binding to autophagy and our inability to reproduce the effect in the GUV system, it is not clear how the Yu et al. data relate to ours or to autophagy, and we thought it best to remove the discussion of the Yu paper.

**Author response image 1. sa2fig1:** NDP52 binds weakly to PA-containing GUVs. Binding is unaffected by mutation of the putative PA binding sequence of Yu et al.

Reviewer #3:The authors dissect protein-protein interactions within the NDP52-ULK1 complex and its membrane interaction using HDX and identify a membrane-interacting region in the coiled-coil domain of its component FIP200. They show the ULK1 complex is recruited to GUV membranes through the identified membrane-binding region, and this membrane recruitment is activated by NDP52. Because NDP52 binds to another region in the coiled-coil domain of FIP200, they conclude that NDP52 allosterically activates the membrane-binding region of the FIP200. This in vitro study is performed carefully, and the presented data appear to be of high quality. Identification of the membrane-interacting region of FIP200 is valuable for the autophagy field and would help future work toward the elucidation of the role of FIP200 in autophagy. However, it seems that the allosteric activation is not fully supported by the data and the work struggles to establish the biological significance of this membrane-binding activity by FIP200.1) It is not clear how the FIP200-mediated membrane association fits in the biological event (selective autophagy). There seem to be several candidate membranes that FIP200 might associate with: the membrane of the targets (bacteria/mitochondria), the ER, the omegasome, the autophagosome/phagophore. However, which of these is the bona fide membrane target of FIP200 is not clear, making it difficult to relate the findings of this work to the mechanism of selective autophagy. NDP52 is also shown to be capable of associating with membranes, which complicates the interpretation. How are the roles of these membrane associations by FIP200 and NDP52 related?

The membrane source for autophagosome biogenesis is a long-standing question in the field, but is not the focus of this study. Most thinking in the field is that the ER is the donor in most cases, but that in some situations there can be other sources. We have used synthetic lipid mixtures that are intended to mimic the ER. However, the increase in generic membrane affinity should contribute regardless of the specific organelle, so these findings should be relevant to any FIP200-dependent form of autophagy even if the membrane source were not the ER.

The strong MR phenotype shown with improved signal to noise in the revised Figure 5 emphasizes that the FIP200 MR is predominantly driving membrane binding, with smaller contributions from NDP52 and probably from the FIP200 NTD.

2) The coiled-coil domain is suggested to associate with membranes through hydrophobic residues exposed upon NDP52 binding. This is a very interesting and thoughtful model deduced from a careful interpretation of the HDX data. However, as hydrophobic residues could bind to many things, this explanation alone is probably not sufficient to convince that the observed membrane-binding is biological. More detailed mechanism of the membrane association could help clarify this. Is the MR predicted to form a typical parallel coiled-coil dimer? If so, would a structural model provide insights into the mechanism of membrane binding?

We think HDX is the gold standard experiment for studying helix melting events, which this appears to be. No better experiment is obvious to us. We have tried to be clearer about the model in the new Figure 6.

3) The authors suggest that NDP52 allosterically activates the membrane binding by FIP200. However, FIP200 is suggested to have weak and intrinsic membrane-binding activity (in the absence of NDP52) (Figure 3D). The overall membrane binding could be enhanced through multivalent FIP200-membrane interactions created by GST-4xUb-mediated clustering. Thus, it seems that the membrane-binding data of the NDP52-ULK1 complex can be explained without invoking such allosteric activation. The NDP52-membrane interactions would also become multivalent in the presence of GST-4xUb so that they would also contribute to the membrane recruitment of the overall complex. Thus, with two membrane-binding domains and clustering, it seems to be not straightforward to determine the magnitude of allosteric activation in the presented data.

We agree with the reviewer that clustering and conformational changes likely work together to increase membrane affinity, as depicted in the new Figure 6.

4) NDP52 alone does not seem to recruit FIP200 to membranes. However, NDP52 alone is shown to be capable of destabilizing the MR. GST-NDP52 enhances the FIP200 membrane association but only modestly. Thus, these data seem to suggest that the membrane-binding affinity of "activated MR" is very low. If hydrophobic residues in such a long region of the coiled-coil were exposed to interact with membranes, as suggested, shouldn't the affinity be higher?

In HDX assays, NDP52 and ULK1 complex are at 16 and 2 μM concentration respectively, so as to saturate binding and ensure we are studying the bound form. In the GUV assays, the ULK1 complex is at 100 nM concentration, to avoid saturating the imaging and lipid binding in the absence of activation. The destabilization of the MR by NDP52 in the absence of lipids is thus observed at a 20-fold higher concentration as compared to the GUV imaging experiment. Since binding to GUVs appears saturating at 100 nM in the presence of NDP52 (but not absence) it would appear that the activated form does in fact bind tightly to membranes.

5) Direct evidence for the allosteric activation is lacking. While structural information is limited to at a low-resolution, the membrane-binding region and NDP52 binding site are both in the coiled-coil. Thus, it may be possible to design and create a mutant FIP200 that can bind to NDB52 but not to membranes without touching MR and NDP52 binding sites directly. Such a mutant protein will be strong support for the suggested allostery.

We are not sure what this mutation would be. It would certainly be nice to have such a mutant.

6) Is the mechanism by which NDP52 associates with membranes known? The authors should create an NDP52 construct that can bind to FIP200 but not to membranes and use it in their ULK1/FIP200 membrane recruitment assays to remove the contribution of NDP52-membrane association.

See the Author response image 1. We constructed what had been reported to be such a mutant based on Yu et al. Cell Research (2019), however the weak binding was unaffected by the putative PA binding motif in this paper. To delve further into the question of whether NDP52 actually binds membranes on its own, and if so, how, would in our view be a different study, one focused on NDP52, as opposed to FIP200, and so beyond the scope of the present work.

The re-analysis of the MR mutant GUV data in Figure 5, showing a strong phenotype with improved signal to noise, should hopefully help allay the reviewer's concern. Clearly the MR is the single largest driver for membrane affinity. The critical roles of NDP52 are to target FIP200 to sites of autophagy initiation and to trigger the conformational change in FIP200.

7) Full-length FIP200 should be included in the sedimentation assay in Figure 3D as it would clarify whether the full-length and the truncated proteins are different in membrane binding.

Full length FIP200 aggregates easily and is found in the pellet after ultracentrifugation even in the absence of lipids, making this experiment difficult to interpret.

8) An MR-only construct should be tested for membrane binding using sedimentation assays and GUV assays to clarify whether the activity of the MR is intrinsically suppressed or not. The MR is said to be not electropositive, but in Figure 3, the C-terminal FIP200 construct shows a better binding with negatively charged liposomes. An MR-only construct might reveal the differences between the MR and the entire CTD. Is the CTD overall electropositive?

Yes, the CTD overall is electropositive. A detailed characterization of the membrane binding properties of the isolated MR fragment would be potentially interesting but it is somewhat beyond the scope of this current paper. The main innovation in the current study is the biochemistry of full-length FIP200 and the complex ULK1 complex, which has been difficult to study in the past because it has been so hard to express in quantity. The conclusions of the study are fully supported we believe without this extension.

9) The oligomeric state of NDP52 should be examined as it has a coiled-coil domain that may oligomerize.

We would be concerned about changing the scope and focus of the paper from FIP200 to NDP52 and have therefore refrained from adding these data, which we do not feel are needed to support the conclusions. This would be interesting as a future direction in a research program more focused on the autophagy adaptors as opposed to the core complexes.

10) The KD between NDP52 and FIP200 coiled-coil would be helpful for understanding the contribution of FIP200 in membrane association if available.

These data are not available in our lab. This is another experiment that would add useful data to the field, but the focus on biochemistry of subcomponents of the system is in our view a sidelight that distracts from the main message and innovation in the study.

11) It may be helpful to present the raw HDX protection profiles (not differences between two samples) as supplements to show protection levels of each samples. Are the coiled-coil regions well protected in the absence of NDP52/liposomes?

These data are now in Figure 1—figure supplement 2B and Figure 2—figure supplement 2A-B. The coiled-coil region was not well protected in the absence of NDP52 or liposomes, and they showed moderate level of exchange rate. Thus these regions seem to have a moderate inherent stability, facilitating their ability to melt and bind membranes when activated.